# TSRM: A LIGHTWEIGHT ARCHITECTURE BASED ON TEMPORAL FEATURE ENCODING FOR TIME SERIES FORECASTING AND IMPUTATION

## ABSTRACT

We introduce a multilayered representation learning architecture called Time Series Representation Model (TSRM) for multivariate time series forecasting and imputation. The architecture is structured around hierarchically ordered encoding layers, each dedicated to an independent representation learning task. Each encoding layer contains a representation layer designed to capture diverse temporal patterns and an aggregation layer responsible for combining the learned representations. The architecture is fundamentally based on a Transformer encoder-like configuration, with self-attention mechanisms at its core. The TSRM architecture outperforms state-of-the-art approaches on most of the seven established benchmark datasets considered in our empirical evaluation for both forecasting and imputation tasks while significantly reducing complexity in the form of learnable parameters. The source code is available at `https://anonymous.4open.science/r/TSRM-D7BE`.

## 1 INTRODUCTION

Time series analysis has high potential in both science and industry. It comprises various disciplines, including time series forecasting, classification, and imputation. By analyzing time series data, we can gain deeper insights into various systems, such as sensor networks (Papadimitriou & Yu, 2006), finance (Zhu & Shasha, 2002), and biological systems like the human body (Ek et al., 2023).

Time series (TS) data often exhibit high dimensionality, with relationships between data points governed by both temporal order and attribute-level structure. Typically, TS are recorded continuously, capturing only a few scalar values at each time step. As single time points usually lack sufficient semantic information for in-depth analysis, research primarily emphasizes temporal variations. These variations offer richer insights into the intrinsic properties of TS, such as continuity and intricate temporal patterns. Since multiple overlapping variations can exist simultaneously, such modeling of temporal dynamics is particularly challenging.

Despite advancements in methodologies to tackle those challenges, such as Recurrent Neural Networks (RNNs) and Convolutional Neural Networks (CNNs), high dimensionality and vanishing/exploding gradients persist, restricting the information flow over long sequences as observed by Hochreiter et al. (2001). With the work of Vaswani et al. (2017), the Transformer architecture was proposed and soon applied in the domain of TS analysis (Wu et al., 2020a). However, due to its design for the domain of Natural Language Processing (NLP) and the resulting use of the point-wise attention mechanism on word embeddings, approaches based on this architecture could not adequately capture all relevant TS characteristics, and they suffered from high computational and memory demands with long-term sequences (Huang et al., 2018; Povey et al., 2018). With time, more specialized implementations emerged. TS forecasting started with improvements to the basic Transformer architecture to overcome the memory bottleneck (Li et al., 2019) with sparse-attention concepts, followed by various enhancements, especially further modifications to the attention part. Examples include Informer (Zhou et al., 2021), Autoformer (Wu et al., 2021), and FEDFormer (Zhou et al., 2022). Transformer-based TS imputation has evolved around hybrid combinations of transformer components with CNN, RNN, auto-encoder, or GAN concepts (Cao et al., 2018; Fortuin et al., 2020; Luo et al., 2018), with recent successes such as SAITS (Du et al., 2023). Despite these sophisticated approaches, the recent work of Zeng et al. (2023) presented a simple

linear model that outperforms all previous models on a number of benchmarks, thus fundamentally questioning the use of transformer models for time series analysis.

Novel approaches have addressed this challenge by abstracting the input data to exploit the modeling capability of transformers more effectively. Most noteworthy approaches evolved around the concept of patching, where the input sequence is split into subsequences (Nie et al., 2022; Liu et al., 2024; Chen et al., 2024). To improve the modeling of temporal variations, other approaches capture temporal patterns at different abstraction levels to learn representations of a TS (Wu et al., 2022). In this paper, we extend this concept of learning temporal representations by introducing a lightweight and adaptive multidimensional framework with a hierarchical design and high configurability to handle complex temporal variations and be applicable to many datasets. Our approach integrates a temporal representation learning concept within a multilayered encoding model, where each encoding layer features a distinct representation learning module paired with a symmetrically structured aggregation layer. This aggregation layer is designed to reverse the learned representations while aggregating key features of the learned representation. Crucially, our encoding layers are designed to allow independent capturing of representations at a different hierarchical level, restoring the original input dimensions to enable hierarchical stacking of layers independent of the input dimension.

## 2    RELATED WORK

**Transformer-based models.** Since its inception in 2017, Transformer (Vaswani et al., 2017) and its numerous derivatives (Zhou et al., 2021; Wu et al., 2021; Zhou et al., 2022; Nie et al., 2022; Du et al., 2023; Zhang & Yan, 2023; Chen et al., 2024; Zhao et al., 2024; Das et al., 2024; Liu et al., 2024) steadily gained traction and are now a well-established approach to time series modeling. One of the more recent works is PatchTST (Nie et al., 2022), which combines the Transformer encoder with subseries-level patches as input encoding to increase efficiency while demonstrating strong modeling capacity. While PatchTST processes each channel of multivariate TS independently, Crossformer (Zhang & Yan, 2023) captures both temporal and cross-channel dependencies. To this end, the model unravels the input TS into two dimensions and features a novel attention layer to learn both types of dependencies efficiently. Pathformer (Chen et al., 2024) is a multiscale transformer with adaptive dual attention to capture temporal dependencies between TS segments of varying granularity. TSRM, while using classical multi-head self-attention internally, does not utilize the Transformer architecture but an adaptation of the Transformer encoder only, to limit the memory footprint and reduce complexity.

**Self-supervised pretraining.** Splitting the training process into pretraining and fine-tuning allows TS models to learn universal representations that can be later utilized for different downstream tasks (Jiang et al., 2022). One work in this field was proposed by Ekambaram et al. (2023). Their TSMixer is built around an MLP backbone, while our approach employs a convolution- and self-attention-based encoder architecture. SimMTM (Dong et al., 2023) and its successor HiMTM (Zhao et al., 2024) also fall into this category. In contrast, CoST (Woo et al., 2022a) learns disentangled feature representations by discriminating the trend and seasonal components. Lee et al. (2024) recently presented PITS. The pre-training approach is very interesting and promising, both in terms of the performance achieved and the accuracy of the representations learned, but was not investigated in this work. Nevertheless, it is planned to evaluate the architecture presented in this paper in a pretraining context in future work.

**Foundation models.** Similar to self-supervised pretraining, time series foundation models learn universal representations of TS and use them for different downstream tasks (Bommasani et al., 2021). However, they are more powerful in that they pretrain on a cross-domain database to generalize across individual target datasets. In recent years, various approaches have been proposed, including TF-C (Zhang et al., 2022b), TimesNet (Wu et al., 2022), FPT (Zhou et al., 2023), Lag-Llama (Rasul et al., 2023), MOMENT (Goswami et al., 2024), MOIRAI (Woo et al., 2024), and TimesFM (Das et al., 2024). TimesFM and FPT are Transformer-based models. TF-C employs a different embedding stage than TSRM, which is based on time-frequency-consistency and contrastive learning. Where TimesNet analyses temporal variations in the 1D input sequence by unfolding it into two dimensions along multiple periods observed over the time axis, TSRM is designed as multilayered representation architecture and it embeds temporal variations into a one-dimensional vector (separately for each layer). MOIRAI follows a patch-based approach with a masked encoder architecture.

Compared to TSRM, MOMENT differs by using patching, a Transformer encoder directly, and self-supervised pretraining on a wide range of datasets.

**Patch-based models.** Patching is a form of input encoding that divides the time series into subsequences, which can be either overlapping or non-overlapping (Nie et al., 2022; Zhang & Yan, 2023; Ekambaram et al., 2023; Zhou et al., 2023; Das et al., 2024; Lee et al., 2024; Chen et al., 2024; Liu et al., 2024; Goswami et al., 2024; Woo et al., 2024). In the basic form, identical-sized patches are sliced from the input TS and fed as tokens to the model (Nie et al., 2022). Pathformer's multiscale division divides the TS into different temporal resolutions using patches of various, dynamically chosen, sizes. Crossformer computes more complex patches encoding both temporal and cross-channel dependencies. iTransformer (Liu et al., 2024) takes the idea to the extreme, operating on patches covering an entire channel of the input TS each. What sets TSRM apart from previous works is the representation learning with multilayered and multidimensional CNN layers, dynamically learned in a novel representation layer, in order to cover different granularities and enable hierarchical representation learning.

**Few/zero-shot learning.** Few-shot learning refers to the capability of a model to generalize from the data domain it is (pre-)trained on to a new target domain using just a few (zero-shot learning: none) target-training instances (Zhou et al., 2023; Rasul et al., 2023; Das et al., 2024; Lee et al., 2024; Woo et al., 2024). Lag-Llama is based on a decoder-only Transformer architecture that uses lags as covariates and processes only univariate TS, while TSRM uses CNN-extracted feature vectors and can handle multivariate TS. Moreover, Lag-Llama is pretrained on a large corpus of multidomain TS data, while FPT utilizes a pretrained language model like BERT (Devlin et al., 2018) as basis. In this work, however, we do not consider the few/zero-shot setting, but instead, we train and optimize TSRM on each dataset separately, focusing on deriving profound representation models that incorporate knowledge about each specific type of TS.

## 3 METHODOLOGY

To address the above challenges in time series analysis, we propose a Time Series Representation Model (TSRM). It consists of a modular and dynamic architecture to model temporal patterns derived from different periods while keeping a low memory profile, including a remarkably small number of trainable parameters. It consists of multiple stacked encoding layers (*EncLayer*), each equipped with a learnable multidimensional representation learning process to capture multiple temporal variations within the input data. Similar to PatchTST (Nie et al., 2022), we explore a channel-independent approach, where each feature channel is processed independently using the same TSRM backbone. However, recognizing that some time series depend on correlations between features, we also introduce an alternative version of TSRM, called TSRM_IFC, which moves away from the channel-independent approach to facilitate the learning of **i**nter-**f**eature **c**orrelations. Details of this second architectural variant are provided later in this section.

### 3.1 MODEL ARCHITECTURE

The multivariate input sequence $\boldsymbol{x}_1 \ldots \boldsymbol{x}_T$, where $\boldsymbol{x}_i \in \mathbb{R}^F$ and $F$ is the amount of input features, is split into $F$ univariate sequences, where each of them is fed independently into the model according to our channel-independent setting but shares the same TSRM backbone, as illustrated in Figure 1.

Each univariate input sequence $x_{1,f_i} \ldots x_{T,f_i}$, where $i \in \{1, ..., F\}$, undergoes a position-wise operation, which extends the univariate sequence to the dimension $d$, and is then added to a positional embedding and normalized with RevIN (Kim et al., 2021), resulting in the embedded sequence $\boldsymbol{e}_{1,f_i} \ldots \boldsymbol{e}_{T,f_i}$, where $\boldsymbol{e}_{*,f_i} \in \mathbb{R}^d$, as illustrated in Figure 1 (left, in grey). Subsequently, the embedded sequence undergoes processing via $N$ consecutive *EncLayer*s, each tasked with deriving representations, learning and encoding temporal features, aggregating encoded features, and restoring the input dimension. These layers utilize the sequence received from the input TS or the preceding *EncLayer*, enabling a hierarchic representation learning. The output of each *EncLayer* is fed into the next, as well as a residual connection, marked with dotted lines in Figure 1, bridging representation matrices across the *EncLayer*s. This layer stacking and the residual connections facilitate a structured feature extraction, similar to deep CNN frameworks known from computer vision (He et al., 2016a), and follow the information-flow principals from Hochreiter et al. (2001). Following

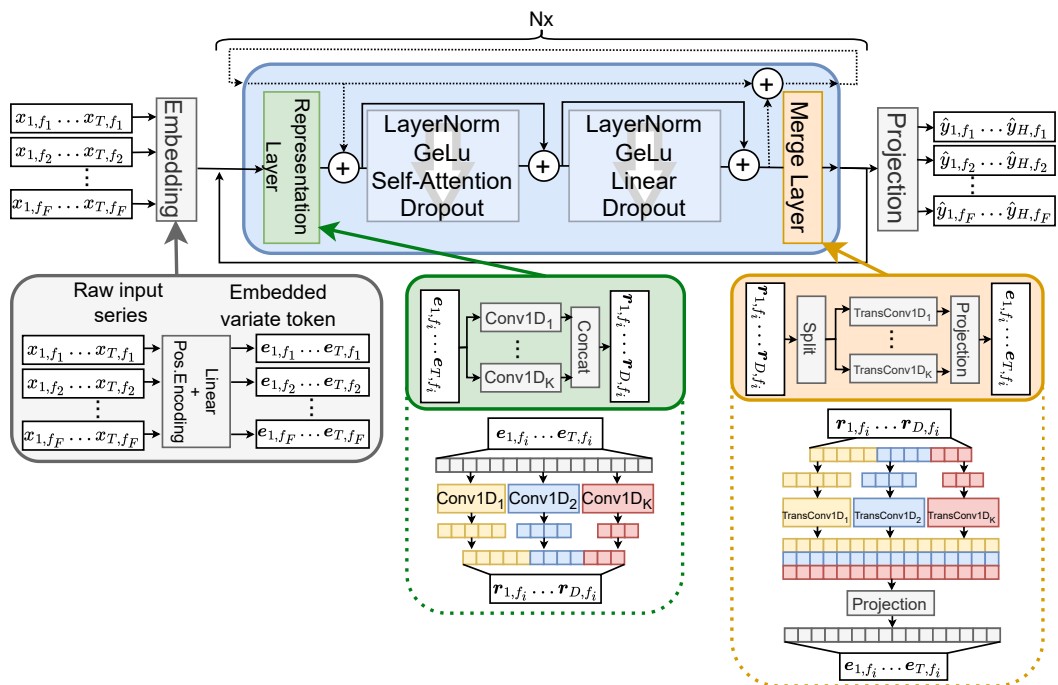

Figure 1: Illustration of the proposed Time Series Representation Model (TSRM) framework, primarily composed of $N$ encoding layers (*EncLayer*s) (upper section in blue), accompanied by the representation layer (*ReprLayer*) (left, in green) and merge layer (*MergeLayer*) (right, in orange).

$N$ *EncLayer*s, we utilize a feed-forward layer to deliver the output sequence, leaned against the concept proven by Das et al. (2023). Finally, we denormalize and reshape the sequence to its original multivariate representation. In the following, we describe each component in detail:

**Encoding Layer (*EncLayer*)** Unlike information-rich word embeddings in NLP, which help models learn language patterns effectively (Selva Birunda & Kanniga Devi, 2021), the informational value of a single point in time is naturally lower, only gaining context information when combined across time steps or the feature dimension. To mitigate this challenge, we introduce the self-attention-based *EncLayer*, which utilizes a potent representation methodology capable of efficiently encapsulating both local and global contextual information. Furthermore, our approach is based on multi-level representation learning, where each *EncLayer* learns a representation of the input sequence, highlights essential patterns, and returns the representation sequence with the dimension of the input, aggregating and embedding essential information. The *EncLayer*, presented in Figure 1 (top in blue), is structured as follows: It starts with the representation layer (*ReprLayer*), tasked with acquiring representations embedded within the sequence and concludes with the merge layer (*MergeLayer*), which aggregates all acquired representations and restores the original input dimension. Hence, the *EncLayer*s are structured to maintain the dimensionality of the input at the output, allowing for $N$ (i.e., the number of *EncLayer* blocks) to be modulated as an independent hyperparameter unrelated to the initial sequence length. Between both layers there are two blocks of layers which are comparable to the Transformer Encoder and have self-attention and linear transformation at their core. Below, we explain each of the two novel layers as well as the two layer blocks in-between in more detail.

**Representation Layer (*ReprLayer*)** The representation layer (*ReprLayer*) is designed to independently learn representations of different abstraction levels from an input sequence and is shown in Figure 1 (bottom in green). Unlike previous approaches (Nie et al., 2022; Liu et al., 2024), which rely on static patches, our method employs a setup of $K$ independent 1D CNN layers with varying kernel sizes, which are designed to capture and integrate representations across different abstrac-

tion levels. Some of the $K$ CNN layers employ small kernels without dilation for identifying basic features, such as sequence details, while others use medium-sized kernels with minimal dilation for intermediate feature recognition, or large kernels with significant dilation for detecting comprehensive features like trends.

To enable a higher level of abstraction, we employ dilation in larger kernels. By default, all kernels are configured with a stride equal to the kernel size to limit the memory footprint. The number of individual CNN layers ($K$), as well as the kernel size and dilation, are hyperparameters and are thus subject to hyperparameter study to find the right constellation for a specific dataset. The outcomes from the $K$ CNN layers are concatenated on the sequence dimension, effectively transforming the input sequence $\boldsymbol{e}_{1,f_i} \ldots \boldsymbol{e}_{T,f_i}$ into the representation $\boldsymbol{r}_{1,f_i} \ldots \boldsymbol{r}_{D,f_i}$, where $D$ corresponds to the total length of the concatenated matrices. This aggregation encapsulates the encoded feature information for each feature $f_i$, spanning varied abstraction levels. The process is comparable to the representation learning of TimesNet (Wu et al., 2022). However, instead of FFT-based capturing and embedding different abstractions into a two-dimensional encoding, we use efficient one-dimensional CNN layers with different abstraction levels and smaller embeddings to enable a low memory profile and fewer trainable parameters, thus reducing the model complexity. The process of the *ReprLayer* for each feature $f_i$ is formalized in Equations 1. The index $f_i$ is omitted for clarity. $\delta_j$ and $s_j$ denote the dilation and kernel size of the $j$-th Conv1D layer with its stride equal to the kernel size and without padding.

$$
\begin{aligned}
\boldsymbol{E} &:= \boldsymbol{e}_1 \ldots \boldsymbol{e}_T, & \boldsymbol{E} &\in \mathbb{R}^{T \times d} \\
\boldsymbol{R}_j &= \text{Conv1D}_j(\boldsymbol{E}), & \boldsymbol{R}_j &\in \mathbb{R}^{D_j \times d}, j \in \{1, \ldots, K\} \\
D_j &= \left\lfloor \frac{T - \delta_j(s_j - 1) - 1}{s_j} + 1 \right\rfloor & & \\
\boldsymbol{R} &= \text{Concat}(\boldsymbol{R}_1, \ldots, \boldsymbol{R}_K), & \boldsymbol{R} &\in \mathbb{R}^{D \times d}, D = \sum_{j=1}^{K} D_j \\
\boldsymbol{r}_1 \ldots \boldsymbol{r}_D &:= \boldsymbol{R} & &
\end{aligned}
\tag{1}
$$

**Merge Layer**    The merge layer (*MergeLayer*), illustrated in Figure 1 (bottom, right in orange) is designed to reverse the dimensional alterations caused by the *ReprLayer* and to aggregate the discovered representations. It comprises $K$ 1D transposed convolution layers utilizing transposed kernels to invert the transformations applied by the corresponding 1D CNN layer of the *ReprLayer*, thereby reinstating the original data dimensions. Therefore, for each feature, the sequence $\boldsymbol{r}_{1,f_i} \ldots \boldsymbol{r}_{D,f_i}$ is segmented in contrast to the concatenation in the *ReprLayer*. These $K$ matrices are then merged using a feed-forward projection, resulting in a sequence that has the exact dimensions of the original input sequence for the *ReprLayer*, $\boldsymbol{e}_{1,f_i} \ldots \boldsymbol{e}_{T,f_i}$. The process of the *MergeLayer* for each feature $f_i$ is formalized in Equations 2. The index $f_i$ is omitted for clarity. The resulting dimensions $D_j$ of the split operation are the same as specified by the 1D CNN layers in the *ReprLayer* in Equations 1.

$$
\begin{aligned}
\boldsymbol{R}_1, \ldots, \boldsymbol{R}_K &:= \text{Split}(\boldsymbol{R}), & \boldsymbol{R} &\in \mathbb{R}^{D \times d}, \boldsymbol{R}_j \in \mathbb{R}^{D_j \times d}, j \in \{1, \ldots, K\} \\
\boldsymbol{R}'_j &= \text{TransConv1D}_j(\boldsymbol{R}_j), & \boldsymbol{R}'_j &\in \mathbb{R}^{T \times d} \\
\boldsymbol{R}' &= \text{Concat}(\boldsymbol{R}'_1, \ldots, \boldsymbol{R}'_K), & \boldsymbol{R}' &\in \mathbb{R}^{T \times dK} \\
\boldsymbol{E} &= \text{FeedForward}_{dK \times d}(\boldsymbol{R}'), & \boldsymbol{E} &\in \mathbb{R}^{T \times d} \\
\boldsymbol{e}_1 \ldots \boldsymbol{e}_T &:= \boldsymbol{E} & &
\end{aligned}
\tag{2}
$$

Situated between the *ReprLayer* and the *MergeLayer*, two blocks of layers facilitate the extraction and amplification of features from the *ReprLayer*. Each is encapsulated within a residual skip connection following the pre-activation design paradigm (He et al., 2016b). The initial block starts with a layer normalization which is succeeded by a GeLu activation function, a multi-head self-attention mechanism, and a dropout operation. The implementation of the multi-head self-attention spans conventional attention akin to the Transformer and sparse self-attention mimicking Wu et al. (2020b). The choice between these two attention mechanisms is adjustable via hyperparameters.

The sequence then advances to the second block, initiating again with a layer normalization, followed by a GeLu activation, a linear layer, and a dropout. The linear layer embedded within this

block differs between the two architecture versions, TSRM and TSRM_IFC (Inter Feature Correlation). In TSRM, the linear layer connects all $d$ dimensions, preserving channel-independence. In the TSRM_IFC, however, it spans all $F \times d$ dimensions, thereby facilitating the learning of inter-feature correlations. The remainder of both architectures maintain feature-separation. Residual connections interlink all of the learned representations across the *EncLayer*s. These connections are added to the resultant matrix from the *ReprLayer*. In addition, the attention and feature-correlation augmented matrix is fused with the residual connection prior to the introduction of the *MergeLayer*, aimed to foster information propagation independent of the *ReprLayer* and the *MergeLayer*.

**Explainability**   Due to the lean and less complex architecture design, it is possible to transfer all attention weights of the *EncLayer*s back and map them to the input time series. This makes it possible to understand the weighting of the attention layer for each of the $N$ *EncLayer* and for each feature separately. This, in turn, permits insight into the functionality and effectiveness of the representation learning process, thus allowing a level of explainability of the model. This proves extremely useful during training on a new dataset but also provides valuable insights about essential patterns in the input sequence for the corresponding task, such as a forecasting. We provide a detailed example in the Appendix A.2.

## 4   EXPERIMENTS

In order to assess the effectiveness of our proposed architecture, we conducted a series of experiments using publicly accessible and well established benchmark datasets from different fields: ECL (Dua et al., 2017), ETT (Zhou et al., 2021) (four subsets: ETTm1, ETTm2, ETTh1, ETTh2), Weather and Exchange (Wu et al., 2021). All datasets were collected from Wu et al. (2021). For more details, please see Section A.1.

### 4.1   EXPERIMENTAL SETUP

For all experiments, we employed early stopping with a threshold of 1% performance increase on the validation set with a patience of three epochs. Hyperparameter tuning was conducted through a random search encompassing various parameters: Count of stacked *EncLayer*s $N$, number of attention heads $h$, encoding dimension $d$, type of attention mechanism (vanilla and sparse; see Section 3.1), and configurations of the *ReprLayer*, including the number of CNN layers, kernel dimensions, dilations, and grouping (see Appendix A.4 for details). Learning rates were initially determined with an automated range test from *LightningAI* (lightning.ai) and adapted during training with a learning rate scheduler (ReduceLROnPlateau) from *PyTorch* with a patience of two epochs. All models were trained with the Adam optimizer on an Nvidia A100 80GB GPU.

### 4.2   LONG-TERM TIME SERIES FORECASTING

Time series forecasting is crucial for predicting future trends and making informed decisions in areas such as finance, healthcare, and supply chain management, enabling better resource planning and risk management.

Given a multivariate input sequence $\boldsymbol{x}_1 \ldots \boldsymbol{x}_T$, where $\boldsymbol{x}_i \in \mathbb{R}^F$ and $F$ is the number of features, the goal is to forecast the prediction sequence $\boldsymbol{y}_1 \ldots \boldsymbol{y}_H$, where $\boldsymbol{y}_i = \boldsymbol{x}_{i+T}$ and $H$ denotes the prediction horizon.

To measure the discrepancy between the prediction sequence $\hat{\boldsymbol{y}}_1 \ldots \hat{\boldsymbol{y}}_H$ and the ground truth $\boldsymbol{y}_1 \ldots \boldsymbol{y}_H$ for a horizon $H$, where $\hat{\boldsymbol{y}}_i, \boldsymbol{y}_i \in \mathbb{R}^F$, we chose the sum of the mean average error (MAE) and mean squared error (MSE) as the loss during training. The loss is calculated and averaged across all $F$ channels and $H$ timesteps to get the overall objective loss:

$$\mathcal{L}_{\text{MAE+MSE}} = \frac{1}{FH} \sum_{i=1}^{H} ||(\hat{\boldsymbol{y}}_i - \boldsymbol{y}_i)||_1 + ||(\hat{\boldsymbol{y}}_i - \boldsymbol{y}_i)||_2^2 \qquad (3)$$

To evaluate the performance of our architecture for long-term TS forecasting, we adopted the procedure of Liu et al. (2024): To support a fair comparison with other approaches, we maintain the input

Table 1: Performance comparison for the multivariate forecasting task with prediction horizons $H \in 96, 192, 336, 720$ and fixed lookback window $T = 96$. Results are averaged over all prediction horizons. Bold/underline indicate best/second.

| Models | TSRM | | TSRM_IFC | | iTransformer | | RLinear | | PatchTST | | Crossformer | | TimesNet | | DLinear | | FEDformer | |
|--------|------|------|------|------|------|------|------|------|------|------|------|------|------|------|------|------|------|------|
| Metrics | MSE | MAE | MSE | MAE | MSE | MAE | MSE | MAE | MSE | MAE | MSE | MAE | MSE | MAE | MSE | MAE | MSE | MAE |
| ECL | 0.193 | 0.277 | 0.193 | 0.277 | **0.178** | **0.270** | 0.219 | 0.298 | 0.205 | 0.290 | 0.244 | 0.334 | 0.193 | 0.295 | 0.212 | 0.300 | 0.214 | 0.327 |
| ETTm1 | **0.381** | **0.392** | 0.386 | 0.394 | 0.407 | 0.410 | 0.414 | 0.408 | 0.387 | 0.400 | 0.513 | 0.495 | 0.400 | 0.406 | 0.403 | 0.407 | 0.448 | 0.452 |
| ETTm2 | 0.277 | 0.323 | **0.275** | **0.320** | 0.288 | 0.332 | 0.286 | 0.327 | 0.281 | 0.326 | 0.757 | 0.611 | 0.291 | 0.333 | 0.350 | 0.401 | 0.305 | 0.349 |
| ETTh1 | **0.438** | **0.431** | 0.438 | 0.437 | 0.454 | 0.448 | 0.446 | 0.434 | 0.469 | 0.455 | 0.529 | 0.522 | 0.458 | 0.450 | 0.456 | 0.452 | 0.440 | 0.460 |
| ETTh2 | 0.375 | 0.398 | **0.368** | **0.396** | 0.383 | 0.407 | 0.374 | 0.399 | 0.387 | 0.407 | 0.942 | 0.684 | 0.414 | 0.427 | 0.559 | 0.515 | 0.437 | 0.449 |
| Weather | 0.243 | 0.268 | **0.240** | **0.263** | 0.258 | 0.278 | 0.272 | 0.291 | 0.259 | 0.281 | 0.259 | 0.315 | 0.259 | 0.287 | 0.265 | 0.317 | 0.309 | 0.360 |
| Exchange | **0.353** | **0.396** | 0.382 | 0.413 | 0.360 | 0.403 | 0.379 | 0.418 | 0.367 | 0.404 | 0.940 | 0.707 | 0.416 | 0.443 | 0.354 | 0.414 | 0.519 | 0.429 |

length for all approaches at $T = 96$, while varying the prediction horizon $H \in \{96, 192, 336, 720\}$, correspondingly. For the evaluation, we consider seven datasets (i.e., ECL, ETTm1, ETTm2, ETTh1, ETTh2, Weather, and Exchange) and compare against multiple state-of-the-art (SOTA) techniques: iTransformer (Liu et al., 2024), RLinear (Li et al., 2023), PatchTST (Nie et al., 2022), Crossformer (Zhang & Yan, 2023), TimesNet (Wu et al., 2022), DLinear (Zeng et al., 2023), and FEDformer (Zhou et al., 2022).

**Results**  The results, presented in Table 1, show a solid performance that matches or outperforms the SOTA approaches, except for the ECL dataset, where our architecture takes the second place, achieving the same MSE as TimesNet but with an improved MAE. On all other investigated datasets, our TSRM or TSRM_IFC architecture achieves the best results on both metrics. It can also be seen that some data sets, such as Weather, achieve better results with TSRM_IFC than with TSRM. This could be because, in this dataset, correlations between individual features are crucial for the prediction, and thus, the TSRM_IFC model performs better, as it allows inter-feature learning. We provide further details with all prediction horizons separated and additional SOTA approaches as comparison in Appendix A.3 and Table 5. SOTA results presented in Tables 1 and 5 were taken from Liu et al. (2024).

## 4.3 IMPUTATION

Time series data from real-world systems often contain missing values, which can arise from sensor malfunctions or environmental conditions. These missing values complicate downstream analysis, necessitating imputation techniques in practical applications. For imputation to provide meaningful replacements for the missing data, the underlying architecture must effectively capture the temporal patterns inherent in the irregularly and partially observed time series.

For the imputation task, a fixed portion $r_m \in [0, 1]$ of values in the original multivariate input sequence is replaced by the masking value $-1$. The positions of replaced values are indicated by random imputation masks $\boldsymbol{m}_1 \ldots \boldsymbol{m}_T$, where $\boldsymbol{m}_i \in \{0, 1\}^F$, $F$ is the number of features and $r_m = \frac{1}{FT} \sum_{i=1}^{T} \boldsymbol{m}_i^\top \boldsymbol{m}_i$. This yields the multivariate masked input sequence $\boldsymbol{x}_1 \ldots \boldsymbol{x}_T$, where $\boldsymbol{x}_i \in \mathbb{R}^F$. The prediction sequence $\hat{\boldsymbol{y}}_1 \ldots \hat{\boldsymbol{y}}_H$ here has the same length as the input sequence, thus $H = T$, and is aimed to accurately reconstruct the original multivariate sequence, that is, the ground truth $\boldsymbol{y}_1 \ldots \boldsymbol{y}_T$.

To measure the discrepancy between the prediction sequence $\hat{\boldsymbol{y}}_1 \ldots \hat{\boldsymbol{y}}_T$ and the ground truth $\boldsymbol{y}_1 \ldots \boldsymbol{y}_T$, where $\hat{\boldsymbol{y}}_i, \boldsymbol{y}_i \in \mathbb{R}^F$, we chose the sum of the mean average error (MAE) and mean squared error (MSE) as the loss during training. For the masked and unmasked regions separately, the losses are calculated and averaged across all $F$ channels and $T$ timesteps. We adjust the contribution of the loss for the masked region, that is $\mathcal{L}_{\text{masked}}$, by weighting factor $\frac{1}{r_m}$. Consequently, we adapt its impact according to the missing ratio. In the following, $\boldsymbol{1}_F$ denotes the 1-vector of length

Table 2: Performance comparison for the multivariate imputation task with missing ratios $r_m \in \{0.125, 0.25, 0.375, 0.5\}$ and a fixed input length of 96. Results are averaged over all missing ratios. Bold/underline indicate best/second.

| Models | TSRM | | TSRM_IFC | | TimesNet | | LightTS | | DLinear | | Stationary | | Autoformer | | Pyraformer | |
|---|---|---|---|---|---|---|---|---|---|---|---|---|---|---|---|---|
| Metrics | MSE | MAE | MSE | MAE | MSE | MAE | MSE | MAE | MSE | MAE | MSE | MAE | MSE | MAE | MSE | MAE |
| ECL | **0.073** | **0.170** | **0.073** | 0.179 | 0.092 | 0.210 | 0.131 | 0.262 | 0.132 | 0.260 | 0.100 | 0.218 | 0.100 | 0.224 | 0.296 | 0.382 |
| ETTm1 | 0.043 | 0.130 | 0.046 | 0.141 | **0.027** | **0.107** | 0.104 | 0.218 | 0.093 | 0.206 | 0.036 | 0.126 | 0.051 | 0.151 | 0.716 | 0.570 |
| ETTm2 | 0.028 | 0.103 | 0.026 | 0.103 | **0.022** | **0.088** | 0.046 | 0.151 | 0.096 | 0.208 | 0.026 | 0.099 | 0.028 | 0.105 | 0.465 | 0.508 |
| ETTh1 | 0.106 | 0.214 | 0.086 | **0.180** | 0.078 | 0.187 | 0.284 | 0.374 | 0.201 | 0.306 | 0.094 | 0.202 | 0.102 | 0.214 | 0.842 | 0.662 |
| ETTh2 | 0.090 | 0.188 | 0.058 | 0.160 | **0.050** | **0.146** | 0.120 | 0.250 | 0.142 | 0.260 | 0.053 | 0.152 | 0.056 | 0.156 | 1.078 | 0.792 |
| Weather | 0.031 | 0.048 | **0.029** | **0.045** | 0.030 | 0.054 | 0.056 | 0.116 | 0.052 | 0.110 | 0.032 | 0.059 | 0.031 | 0.057 | 0.152 | 0.234 |

$F$, that is, $\mathbf{1}_F = (1, ..., 1) \in \mathbb{R}^F$:

$$\mathcal{L}_{\text{masked}} = \frac{1}{r_m F T} \sum_{i=1}^{T} ||\boldsymbol{m}_i \odot (\hat{\boldsymbol{y}}_i - \boldsymbol{y}_i)||_1 + ||\boldsymbol{m}_i \odot (\hat{\boldsymbol{y}}_i - \boldsymbol{y}_i)||_2^2$$

$$\mathcal{L}_{\text{unmasked}} = \frac{1}{(1-r_m) F T} \sum_{i=1}^{T} ||(\mathbf{1}_F - \boldsymbol{m}_i) \odot (\hat{\boldsymbol{y}}_i - \boldsymbol{y}_i)||_1 + ||(\mathbf{1}_F - \boldsymbol{m}_i) \odot (\hat{\boldsymbol{y}}_i - \boldsymbol{y}_i)||_2^2 \quad (4)$$

$$\mathcal{L}_{\text{imputation}} = \frac{1}{r_m} \mathcal{L}_{\text{masked}} + \mathcal{L}_{\text{unmasked}}$$

To evaluate the performance of our architecture for TS imputation, we adopted the experimental setup of Wu et al. (2022) and introduced random data omissions into all datasets, resulting in four distinct missing rates: $r_m \in \{12.5\%, 25\%, 37.5\%, 50\%\}$.

For the evaluation, we consider six datasets (i.e., ECL, ETTm1, ETTm2, ETTh1, ETTh2, Weather) and compare against multiple SOTA techniques: TimesNet (Wu et al., 2022), LightTS (Zhang et al., 2022a), DLinear (Zeng et al., 2022), Stationary (Liu et al., 2022b), Autoformer (Wu et al., 2021), and Pyraformer (Liu et al., 2021).

**Results** We present the results of our proposed architecture in Table 2. On the ECL and Weather dataset, our architecture performs considerably well, whereas on the ETT datasets we were not able to match current SOTA results. Despite the good results with 20.65% performance increase on the MSE metric for ECL compared to TimesNet, we report comparable decreases in performance on the ETT datasets. Despite this, our TSRM approach outperforms LightTS, DLinear, and Pyraformer across all datasets for both averaged metrics. Similar to the forecasting experiments' results, the Weather dataset performs better with the TSRM_IFC than with the TSRM, which reinforces the suspicion that inter-feature correlation plays a major role in this dataset's modeling performance. We provide further details with all missing ratios separated and additional SOTA approaches as comparison in Appendix A.5 and Table 8. All reported results in Tables 2 and 8 were taken from Wu et al. (2022).

## 4.4 COMPLEXITY ANALYSIS

Our results show stable results that meet or even exceed SOTA results, especially for TS forecasting. However, in addition to the pure performance metrics, a model's complexity should also play an essential role in assessing an architecture's quality, not least to maintain its economy and applicability in practice. Our architecture is based on simple and less complex layers that require only a comparably small number of trainable parameters. Furthermore, it is designed for memory efficiency to have a smaller memory footprint. For example, a model for the ETTh1 dataset in the prediction task (96/96) with 8 batches requires only 500 MB of GPU memory in total. In Table 3, we compare the number of trainable parameters of SOTA architectures most similar to ours as an indicator for complexity. The values shown for the SOTA approaches originate from the work of Wang et al. (2024), the Time-Series-Library[1], and were collected separately for all datasets and the corresponding configuration. All reported trainable parameters of TSRM were collected from the corresponding TS forecasting models reported in Table 1. 96 was always selected as both the lookback and prediction

---

[1]https://github.com/thuml/Time-Series-Library

Table 3: Amount of trainable parameters in million. Lookback window is fixed to 96.

| Model | ETTh1 | ETTm2 | ECL | Exchange | Weather | Average |
|---|---|---|---|---|---|---|
| TimesNet | 0.605 | **1.191** | 150.304 | 4.708 | 1.193 | 31.600 |
| Autoformer | 10.535 | 10.535 | 12.143 | 10.541 | 10.607 | 10.872 |
| Transformer | 10.540 | 10.540 | 10.518 | 10.543 | 10.590 | 10.546 |
| PatchTST | 3.751 | 10.056 | 6.903 | 6.903 | 6.903 | 6.903 |
| Crossformer | 42.176 | 42.139 | 9.261 | 0.437 | **0.123** | 18.827 |
| iTransformer | **0.224** | 4.833 | 4.833 | **0.224** | 4.833 | 2.989 |
| TSRM | 0.857 | 2.781 | **0.161** | 0.382 | 0.338 | **0.904** |

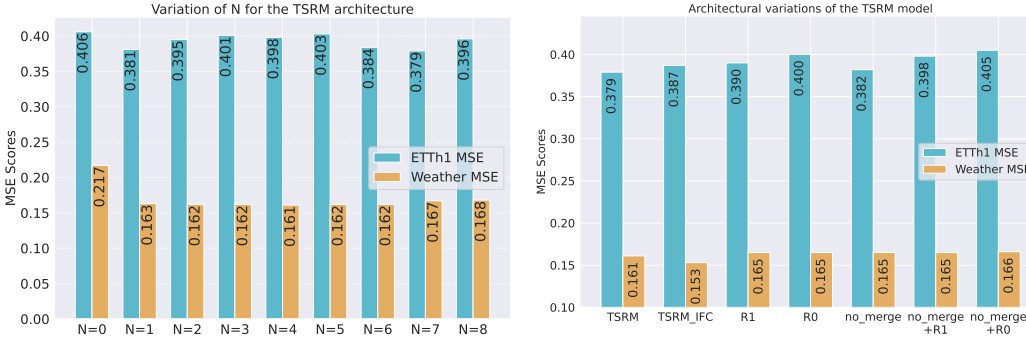

Figure 2: Ablation study results for the architecture variations (right) and sensitivity study for the hyperparameter $N$ (left) on the datasets Weather and ETTh1.

horizon to ensure comparability. The last column shows the average number of parameters across datasets. While most models, on average, have more than 10 million trainable parameters (median $6.903M \pm 27.648M$), TSRM exhibits a much lower complexity with only a few hundred thousand trainable parameters, with the exception of the ETTm2 dataset. However, it is worth noting that despite this low complexity of the TSRM architecture, the DLinear model of Zeng et al. (2022) requires only about 0.018M trainable parameters and still performs very well on most datasets, as shown in Table 1.

## 5  ABLATION STUDY

To evaluate the contribution of different modules within TSRM, we conduct ablation studies centered on our proposed *EncLayer*. Specifically, we examine the effect of varying the number of stacked *EncLayer*s through a sensitivity analysis of $N$. Additionally, to further explore the role of the *EncLayer* in learning time series representations, we modify its configuration to remove the *MergeLayer* (no_merge) and include only a single CNN layer in the *ReprLayer* ($R = 1$). For all experiments, we use ETTh1 as the dataset, which performs well without feature interaction (TSRM), and Weather, which shows better results when used with TSRM_IFC, i.e. with feature interaction. Note that all experiments were carried out with the TSRM variant, not with TSRM_IFC.

**Varying the number of *EncLayer*s.**  We investigate the sensitivity of our architecture towards the amount of *EncLayer*s ($N$). Therefore, we picked the best hyperparamter constellations of the forecasting task and run experiments with $N \in \{0, 1, 2, \ldots 8\}$. Figure 2 (left) shows the results of all variations. The configuration of $N = 0$ means that no *EncLayer* is involved. For both datasets, we observe a plateau of the MSE metric in certain ranges. For example, the ETTh1 dataset performs best with a fairly high number of $N$, while Weather performs best with $N = 4$ and then gets worse as $N$ increases. For both datasets, however, the results are significantly improved with at least one *EncLayer* and with an increasing number the result can be further improved until a plateau is formed. This shows the benefit of stacked *EncLayer*s for these two datasets. The configuration without any *EncLayer* ($N = 0$) yields the worst result on the ETTh1 dataset. However, it is worth noting that despite the lack of any complex logic in the case of $N = 0$, the result is comparable to DLinear on the weather dataset on the ETTh1 dataset. This can be explained by the fact that the remaining architecture is comparable to that of DLinear. More details can be found in the Appendix A.6.

**Architecture variations** We explore the influence of various architectural components on the performance of TSRM. Specifically, we assess the effect of the *MergeLayer* by blocking gradient flow to prevent the layer from learning aggregations in the experiment *no_merge*. As a result, the dimensional changes caused by the transposed CNN layer are preserved to allow further stacking of the encoding layer, but without trainable parameters. Additionally, we reduce $R$ to a single CNN layer with a kernel size of three and a dilation of one in the experiment *R1* and with a deactivated *MergeLayer* in the experiment *R1+no_merge*. In a further step, we restrict the CNN layer to a kernel size of one, eliminating any structural learning effects and limiting its function to position-wise weighting, as tested in the experiment *R0*, and with a deactivated *MergeLayer* in the experiment *no_merge+R0*. Figure 2 (right) shows the results of all ablation experiments. A decline in performance of the TSRM architecture is evident throughout the ablation experiments. The most notable drop occurs between the original TSRM architecture and the *R0* experiment, where a considerable loss in performance highlights the critical role of the CNN layers withing the *ReprLayer*. However, it is important to note that although the performance changes without the trainable merge layer are not significant compared to the original TSRM architecture, the role of the merge layer is mainly to undo the dimensional changes caused by the representation layer and thus allow stacking of the encoding layer regardless of the input length. Reducing the number of 1D convolutions (i.e., $R = 1$) and preventing structural learning with a kernel size of 1 further reduces performance, although the drop is less pronounced than between the scenarios with and without merge layers. Interestingly, the experiment *no_merge+R0*, in which the architectural structure is changed, is very similar to that of iTransformer (Liu et al., 2024) and leads to comparable performance results, as shown in the appendix A.3 (Table 5). The weather dataset shows no performance changes between the experiments *no_merge* and *no_merge+R1*, which can be explained by the fact that the most powerful model in this setup and thus the configuration used in this experiment uses the same CNN configuration as in $R1$, which accordingly leads to no performance changes. More details can be found in the Appendix A.6.

## 6 CONCLUSION

We introduced a new architecture for time series prediction and imputation, the Time Series Representation Model (TSRM). This model uses hierarchically organized encoding layers (*EncLayer*) designed to independently learn representations from the input sequence at different levels of abstraction, with each layer passing learned and aggregated features to the next. The *EncLayer* is largely based on the concept of self-attention and consists of a representation layer and an aggregation layer, which are responsible for representing the input sequence at different levels of abstraction as well as aggregating and embedding the learned or highlighted representations. The architecture is designed to be of low complexity while supporting explainability in the form of detailed attention highlighting. Our empirical evaluation showed that TSRM is able to outperform SOTA approaches on a number of well-established benchmark datasets in the area of time series forecasting and imputation while significantly reducing complexity in form of the number of trainable parameters. In future work, we plan to evaluate the architecture regarding a pretraining/fine-tuning, few/zero-shot learning and foundation model approach, as well as further tasks such as classification and anomaly detection.

## 7 REPRODUCIBILITY STATEMENT

In Section 3, we have strictly formalized the model architecture with equations. The Appendix includes all the implementation details, including dataset descriptions, detailed metrics, and configurations. The source code is available at `https://anonymous.4open.science/r/TSRM-D7BE`.

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

# A  APPENDIX

## A.1  DATASETS

Below, we provide more details on the datasets used in our experiments. Please find a detailed overview of all employed benchmark datasets in Table 4.

**Electricity Load Diagram (ECL)**: The Electricity dataset, available at UCI (Dua et al., 2017), contains electricity consumption data measured in kilowatt-hours (kWh). It includes data from 370 clients collected every 15 minutes for 48 months, starting from January 2011 to December 2014.

**Weather**: The weather dataset contains the recordings of 21 meteorological factors, such as temperature, humidity, and air pressure, collected every 10 minutes from the Weather Station of the Max Planck Biogeochemistry Institute in Jena, Germany in 2020 (Wu et al., 2021).

**Exchange**: This dataset collects the daily exchange rates of 8 different currencies (Australia, British, Canada, Switzerland, China, Japan, New Zealand, and Singapore) from 1990 to 2016 (Wu et al., 2021).

**Electricity Transformer Temperature (ETT)**: The ETT dataset comprises data collected from electricity transformers over a time period from July 1, 2016, to June 26, 2018. ETT consists of 4 subsets, where ETTh1 and ETTh2 contain records with hourly resolution, while ETTm1 and ETTm2 are recorded every 15 minutes. In total, ETT includes 69,680 data points without any missing values. Each record contains seven features, including oil temperature and six different types of external power load features (Zhou et al., 2021).

Table 4: Details of the used benchmark datasets. The assignment to train, validation, or test follows the established procedure (Wu et al., 2021).

| Dataset | Channels | Size (train / val / test) | Frequency | Information |
|---|---|---|---|---|
| ECL | 321 | 18317 / 2633 / 5261 | Hourly | Electricity |
| ETTm1,ETTm2 | 7 | 34465 / 11521 / 11521 | 15min | Electricity |
| ETTh1,ETTh2 | 7 | 8545 / 2881 / 2881 | Hourly | Electricity |
| Exchange | 8 | 5120 / 665 / 1422 | Daily | Economy |
| Weather | 21 | 36792 / 5271 / 10540 | 10min | Weather |

## A.2  EXPLAINABILITY WITH ATTENTION WEIGHT HIGHLIGHTING

Our methodology's fundamental architectural principle is predicated on utilizing the attention mechanism as its central component and maintaining dimensional consistency throughout all *EncLayer*s. As detailed in Section 3, the attention layers play a pivotal role in enhancing the representations from the representation layers.This approach, combined with the low complexity of our architecture, enables us to extract and investigate the attention weights of all *EncLayer*s, offering valuable insights into our architecture's functioning and decision making. Due to its design, we are able to extract and analyze individual attention weights for all $N$ *EncLayer*s' attention layers and all $F$ features individually. This means that separate attention weights can be generated for each feature and *EncLayer*, thereby enabling their analysis in isolation and combined. For this, it is imperative to revert the matrix dimensions dictated by the *ReprLayer*s back to those of the input TS. This transformation employs the identical transpose CNN layer utilized in the *MergeLayer*s, albeit with static weight matrices, designed to calculate the mean attention weight for each value. The $N$ back-transformed attention matrices can then be visualized together with the output sequence to analyze the architecture's weighting during imputation, or prediction. This can be done for all $N$ *EncLayer*s and features individually or as a sum over all *EncLayer*s to get an overview of all weights.

Figure 3 shows an example with the ETTh1 dataset on the first feature during a forecasting task with all three *EncLayer* separately visualized, including the combined attention weights at the bottom. The solid green line represents the initial input series, followed by a dotted blue line after the 96th value. This dotted blue trajectory delineates the target horizon. The red line indicates the prediction of the model. The emphasis of attention is subject to a threshold value of 0.85 (normalized) in order to emphasize only the most important aspects of attention. We can observe that the attention from the first *EncLayer* is more distributed and mainly focuses on high and low points in the sequence.

Table 5: Performance comparison for the multivariate forecasting task with prediction horizons $H \in 96, 192, 336, 720$ and fixed lookback window $T = 96$. AVG shows the averaged result over all prediction horizons per dataset and model. Bold/underline indicate best/second.

| Models | | TSRM | | TSRM_IFC | | iTransformer | | RLinear | | PatchTST | | Crossformer | | TiDE | | TimesNet | | DLinear | | SCINet | | FEDformer | | Stationary | | Autoformer | |
|---|---|---|---|---|---|---|---|---|---|---|---|---|---|---|---|---|---|---|---|---|---|---|---|---|---|---|---|
| | H | MSE | MAE | MSE | MAE | MSE | MAE | MSE | MAE | MSE | MAE | MSE | MAE | MSE | MAE | MSE | MAE | MSE | MAE | MSE | MAE | MSE | MAE | MSE | MAE | MSE | MAE |
| ECL | 96 | 0.168 | 0.255 | 0.168 | 0.255 | 0.148 | 0.240 | 0.201 | 0.281 | 0.181 | 0.270 | 0.219 | 0.314 | 0.237 | 0.329 | 0.168 | 0.272 | 0.197 | 0.282 | 0.247 | 0.345 | 0.193 | 0.308 | 0.169 | 0.273 | 0.201 | 0.317 |
| | 192 | 0.176 | 0.262 | 0.176 | 0.262 | 0.162 | 0.253 | 0.201 | 0.283 | 0.188 | 0.274 | 0.231 | 0.322 | 0.236 | 0.330 | 0.184 | 0.289 | 0.196 | 0.285 | 0.257 | 0.355 | 0.201 | 0.315 | 0.182 | 0.286 | 0.222 | 0.334 |
| | 336 | 0.192 | 0.278 | 0.192 | 0.278 | 0.178 | 0.269 | 0.215 | 0.298 | 0.204 | 0.293 | 0.246 | 0.337 | 0.249 | 0.344 | 0.198 | 0.300 | 0.209 | 0.301 | 0.269 | 0.369 | 0.214 | 0.329 | 0.200 | 0.304 | 0.231 | 0.338 |
| | 720 | 0.234 | 0.312 | 0.234 | 0.312 | 0.225 | 0.317 | 0.257 | 0.331 | 0.246 | 0.324 | 0.280 | 0.363 | 0.284 | 0.373 | 0.220 | 0.320 | 0.245 | 0.333 | 0.299 | 0.390 | 0.246 | 0.355 | 0.222 | 0.321 | 0.254 | 0.361 |
| | AVG | 0.193 | 0.277 | 0.193 | 0.277 | 0.178 | 0.270 | 0.219 | 0.298 | 0.205 | 0.290 | 0.244 | 0.334 | 0.252 | 0.344 | 0.193 | 0.295 | 0.212 | 0.300 | 0.268 | 0.365 | 0.214 | 0.327 | 0.193 | 0.296 | 0.227 | 0.338 |
| ETTm1 | 96 | 0.314 | 0.352 | 0.314 | 0.354 | 0.334 | 0.368 | 0.355 | 0.376 | 0.329 | 0.367 | 0.404 | 0.426 | 0.364 | 0.387 | 0.338 | 0.375 | 0.345 | 0.372 | 0.418 | 0.438 | 0.379 | 0.419 | 0.386 | 0.398 | 0.505 | 0.475 |
| | 192 | 0.367 | 0.379 | 0.369 | 0.383 | 0.377 | 0.391 | 0.391 | 0.392 | 0.367 | 0.385 | 0.450 | 0.451 | 0.398 | 0.404 | 0.374 | 0.387 | 0.380 | 0.389 | 0.439 | 0.450 | 0.426 | 0.441 | 0.459 | 0.444 | 0.553 | 0.496 |
| | 336 | 0.393 | 0.401 | 0.398 | 0.403 | 0.426 | 0.420 | 0.424 | 0.415 | 0.399 | 0.410 | 0.532 | 0.515 | 0.428 | 0.425 | 0.410 | 0.411 | 0.413 | 0.413 | 0.490 | 0.485 | 0.445 | 0.459 | 0.495 | 0.464 | 0.621 | 0.537 |
| | 720 | 0.448 | 0.435 | 0.461 | 0.437 | 0.491 | 0.459 | 0.487 | 0.450 | 0.454 | 0.439 | 0.666 | 0.589 | 0.487 | 0.461 | 0.478 | 0.450 | 0.474 | 0.453 | 0.595 | 0.550 | 0.543 | 0.490 | 0.585 | 0.516 | 0.671 | 0.561 |
| | AVG | 0.381 | 0.392 | 0.386 | 0.394 | 0.407 | 0.410 | 0.414 | 0.408 | 0.387 | 0.400 | 0.513 | 0.495 | 0.419 | 0.419 | 0.400 | 0.406 | 0.403 | 0.407 | 0.486 | 0.481 | 0.448 | 0.452 | 0.481 | 0.456 | 0.588 | 0.517 |
| ETTm2 | 96 | 0.173 | 0.257 | 0.169 | 0.253 | 0.180 | 0.264 | 0.182 | 0.265 | 0.175 | 0.259 | 0.287 | 0.366 | 0.207 | 0.305 | 0.187 | 0.267 | 0.193 | 0.292 | 0.286 | 0.377 | 0.203 | 0.287 | 0.192 | 0.274 | 0.255 | 0.339 |
| | 192 | 0.239 | 0.301 | 0.236 | 0.297 | 0.250 | 0.309 | 0.246 | 0.304 | 0.241 | 0.302 | 0.414 | 0.492 | 0.290 | 0.364 | 0.249 | 0.309 | 0.284 | 0.362 | 0.399 | 0.445 | 0.269 | 0.328 | 0.280 | 0.339 | 0.281 | 0.340 |
| | 336 | 0.297 | 0.338 | 0.292 | 0.332 | 0.311 | 0.348 | 0.307 | 0.342 | 0.305 | 0.343 | 0.597 | 0.542 | 0.377 | 0.422 | 0.321 | 0.351 | 0.369 | 0.427 | 0.637 | 0.591 | 0.325 | 0.366 | 0.334 | 0.361 | 0.339 | 0.372 |
| | 720 | 0.398 | 0.397 | 0.404 | 0.397 | 0.412 | 0.407 | 0.407 | 0.398 | 0.402 | 0.400 | 1.730 | 1.042 | 0.558 | 0.524 | 0.408 | 0.403 | 0.554 | 0.522 | 0.960 | 0.735 | 0.421 | 0.415 | 0.417 | 0.413 | 0.433 | 0.432 |
| | AVG | 0.277 | 0.323 | 0.275 | 0.32 | 0.288 | 0.332 | 0.286 | 0.327 | 0.281 | 0.326 | 0.757 | 0.611 | 0.358 | 0.404 | 0.291 | 0.333 | 0.350 | 0.401 | 0.571 | 0.537 | 0.305 | 0.349 | 0.306 | 0.347 | 0.327 | 0.371 |
| ETTh1 | 96 | 0.379 | 0.394 | 0.387 | 0.401 | 0.386 | 0.405 | 0.386 | 0.395 | 0.414 | 0.419 | 0.423 | 0.448 | 0.479 | 0.464 | 0.384 | 0.402 | 0.386 | 0.400 | 0.654 | 0.599 | 0.376 | 0.419 | 0.513 | 0.491 | 0.449 | 0.459 |
| | 192 | 0.433 | 0.425 | 0.426 | 0.428 | 0.441 | 0.436 | 0.437 | 0.424 | 0.460 | 0.445 | 0.471 | 0.474 | 0.525 | 0.492 | 0.436 | 0.429 | 0.437 | 0.432 | 0.719 | 0.631 | 0.420 | 0.448 | 0.534 | 0.504 | 0.500 | 0.482 |
| | 336 | 0.464 | 0.446 | 0.463 | 0.451 | 0.487 | 0.458 | 0.479 | 0.446 | 0.501 | 0.466 | 0.570 | 0.546 | 0.565 | 0.515 | 0.491 | 0.469 | 0.481 | 0.459 | 0.778 | 0.659 | 0.459 | 0.465 | 0.588 | 0.535 | 0.521 | 0.496 |
| | 720 | 0.474 | 0.459 | 0.477 | 0.466 | 0.503 | 0.491 | 0.481 | 0.470 | 0.500 | 0.488 | 0.653 | 0.621 | 0.594 | 0.558 | 0.521 | 0.500 | 0.519 | 0.516 | 0.836 | 0.699 | 0.506 | 0.507 | 0.643 | 0.616 | 0.514 | 0.512 |
| | AVG | 0.438 | 0.431 | 0.438 | 0.437 | 0.454 | 0.448 | 0.446 | 0.434 | 0.469 | 0.455 | 0.529 | 0.522 | 0.541 | 0.507 | 0.458 | 0.450 | 0.456 | 0.452 | 0.747 | 0.647 | 0.440 | 0.460 | 0.570 | 0.537 | 0.496 | 0.487 |
| ETTh2 | 96 | 0.286 | 0.334 | 0.296 | 0.345 | 0.297 | 0.349 | 0.288 | 0.338 | 0.302 | 0.348 | 0.745 | 0.584 | 0.400 | 0.440 | 0.340 | 0.374 | 0.333 | 0.387 | 0.707 | 0.621 | 0.358 | 0.397 | 0.476 | 0.458 | 0.346 | 0.388 |
| | 192 | 0.370 | 0.388 | 0.371 | 0.391 | 0.380 | 0.400 | 0.374 | 0.390 | 0.388 | 0.400 | 0.877 | 0.656 | 0.528 | 0.509 | 0.402 | 0.414 | 0.477 | 0.476 | 0.860 | 0.689 | 0.429 | 0.439 | 0.512 | 0.493 | 0.456 | 0.452 |
| | 336 | 0.417 | 0.428 | 0.405 | 0.420 | 0.428 | 0.432 | 0.415 | 0.426 | 0.426 | 0.433 | 1.043 | 0.731 | 0.643 | 0.571 | 0.452 | 0.452 | 0.594 | 0.541 | 1.000 | 0.744 | 0.496 | 0.487 | 0.552 | 0.551 | 0.482 | 0.486 |
| | 720 | 0.427 | 0.443 | 0.399 | 0.426 | 0.427 | 0.445 | 0.420 | 0.440 | 0.431 | 0.446 | 1.104 | 0.763 | 0.874 | 0.679 | 0.462 | 0.468 | 0.831 | 0.657 | 1.249 | 0.838 | 0.463 | 0.474 | 0.562 | 0.560 | 0.515 | 0.511 |
| | AVG | 0.375 | 0.398 | 0.368 | 0.396 | 0.383 | 0.407 | 0.374 | 0.399 | 0.387 | 0.407 | 0.942 | 0.684 | 0.611 | 0.550 | 0.414 | 0.427 | 0.559 | 0.515 | 0.954 | 0.723 | 0.437 | 0.449 | 0.526 | 0.516 | 0.450 | 0.459 |
| Weather | 96 | 0.161 | 0.202 | 0.153 | 0.200 | 0.174 | 0.214 | 0.192 | 0.232 | 0.177 | 0.218 | 0.158 | 0.230 | 0.202 | 0.261 | 0.172 | 0.220 | 0.196 | 0.255 | 0.221 | 0.306 | 0.217 | 0.296 | 0.173 | 0.223 | 0.266 | 0.336 |
| | 192 | 0.207 | 0.245 | 0.202 | 0.245 | 0.221 | 0.254 | 0.240 | 0.271 | 0.225 | 0.259 | 0.206 | 0.277 | 0.242 | 0.298 | 0.219 | 0.261 | 0.237 | 0.296 | 0.261 | 0.340 | 0.276 | 0.336 | 0.245 | 0.285 | 0.307 | 0.367 |
| | 336 | 0.261 | 0.285 | 0.264 | 0.268 | 0.278 | 0.296 | 0.292 | 0.307 | 0.278 | 0.297 | 0.272 | 0.335 | 0.287 | 0.335 | 0.280 | 0.306 | 0.283 | 0.335 | 0.309 | 0.378 | 0.339 | 0.380 | 0.321 | 0.338 | 0.359 | 0.395 |
| | 720 | 0.343 | 0.339 | 0.342 | 0.34 | 0.358 | 0.347 | 0.364 | 0.353 | 0.354 | 0.348 | 0.398 | 0.418 | 0.351 | 0.386 | 0.365 | 0.359 | 0.345 | 0.381 | 0.377 | 0.427 | 0.403 | 0.428 | 0.414 | 0.410 | 0.419 | 0.428 |
| | AVG | 0.243 | 0.268 | 0.240 | 0.263 | 0.258 | 0.278 | 0.272 | 0.291 | 0.259 | 0.281 | 0.259 | 0.315 | 0.271 | 0.320 | 0.259 | 0.287 | 0.265 | 0.317 | 0.292 | 0.363 | 0.309 | 0.360 | 0.288 | 0.314 | 0.338 | 0.382 |
| Exchange | 96 | 0.080 | 0.198 | 0.088 | 0.208 | 0.086 | 0.206 | 0.093 | 0.217 | 0.088 | 0.205 | 0.256 | 0.367 | 0.094 | 0.218 | 0.107 | 0.234 | 0.088 | 0.218 | 0.267 | 0.396 | 0.148 | 0.278 | 0.111 | 0.237 | 0.197 | 0.323 |
| | 192 | 0.168 | 0.291 | 0.179 | 0.307 | 0.177 | 0.299 | 0.184 | 0.307 | 0.176 | 0.299 | 0.470 | 0.509 | 0.184 | 0.307 | 0.226 | 0.344 | 0.176 | 0.315 | 0.351 | 0.459 | 0.271 | 0.315 | 0.219 | 0.335 | 0.300 | 0.369 |
| | 336 | 0.315 | 0.406 | 0.381 | 0.439 | 0.331 | 0.417 | 0.351 | 0.432 | 0.301 | 0.397 | 1.268 | 0.883 | 0.349 | 0.431 | 0.367 | 0.448 | 0.313 | 0.427 | 1.324 | 0.853 | 0.460 | 0.427 | 0.421 | 0.476 | 0.509 | 0.524 |
| | 720 | 0.849 | 0.688 | 0.879 | 0.698 | 0.847 | 0.691 | 0.886 | 0.714 | 0.901 | 0.714 | 1.767 | 1.068 | 0.852 | 0.698 | 0.964 | 0.746 | 0.839 | 0.695 | 1.058 | 0.797 | 1.195 | 0.695 | 1.092 | 0.769 | 1.447 | 0.941 |
| | AVG | 0.353 | 0.396 | 0.382 | 0.413 | 0.360 | 0.403 | 0.379 | 0.418 | 0.367 | 0.404 | 0.940 | 0.707 | 0.370 | 0.414 | 0.416 | 0.443 | 0.354 | 0.414 | 0.750 | 0.626 | 0.519 | 0.429 | 0.461 | 0.454 | 0.613 | 0.539 |

Later, attention seems to be more dense and switches on selected representatives of repeating sub-areas in the time series, e.g. the pattern around 75, as well as on the last known value before the horizon, giving the impression that it is focusing on these areas more closely. However, it should be noted that this is not an evaluation but rather an interpretation of a snapshot, and cannot be taken as evidence of true explainability.

## A.3 FORECASTING

We extend our evaluation from Section 4.2 with a more detailed evaluation, including more SOTA approaches for the comparison, and we provide the results of all prediction lengths separately. For the evaluation, we consider seven datasets (i.e., ECL, ETTm1, ETTm2, ETTh1, ETTh2, Weather, and Exchange) and compare against multiple SOTA techniques: iTransformer (Liu et al., 2024), RLinear (Li et al., 2023), PatchTST (Nie et al., 2022), Crossformer (Zhang & Yan, 2023), TiDE (Das et al., 2023), TimesNet (Wu et al., 2022), DLinear (Zeng et al., 2023), SCINet (Liu et al., 2022a), FEDformer (Zhou et al., 2022), Stationary (Liu et al., 2022b), and Autoformer (Wu et al., 2021).

Table 5 shows the extended results with all four prediction horizons ($H$) separated. We further report the training and inference time for the ETTh1 dataset with the best configuration (see table 6) and a horizon of 96 with 28 seconds/epoch for training and 6 seconds/epoch for inference. For the weather dataset, also with the best configuration and a horizon of 96, we report 194 seconds/epoch during training and 52 seconds/epoch during inference.

To enhance the reproducibility of our experiments, we also report the best hyperparameter configurations for the TSRM architecture for the forecasting task. Table 6 shows all configurations for all prediction horizons.

**Multiple random runs** To evaluate the significance of our forecasting experiments, we provide the mean and standard deviation (STD) for two exemplary datasets, ETTh2 and Weather, in Table 7. To calculate the mean and the STD for MSE and MAE, we performed all runs five times with different seeds.

## A.4 EXPERIMENTAL DETAILS

This section gives additional details about our forecasting and imputation related experiments. All experiments were part of a hyperparameter study utilizing a random search methodology. We examined the subsequent hyperparameters within these specified ranges:

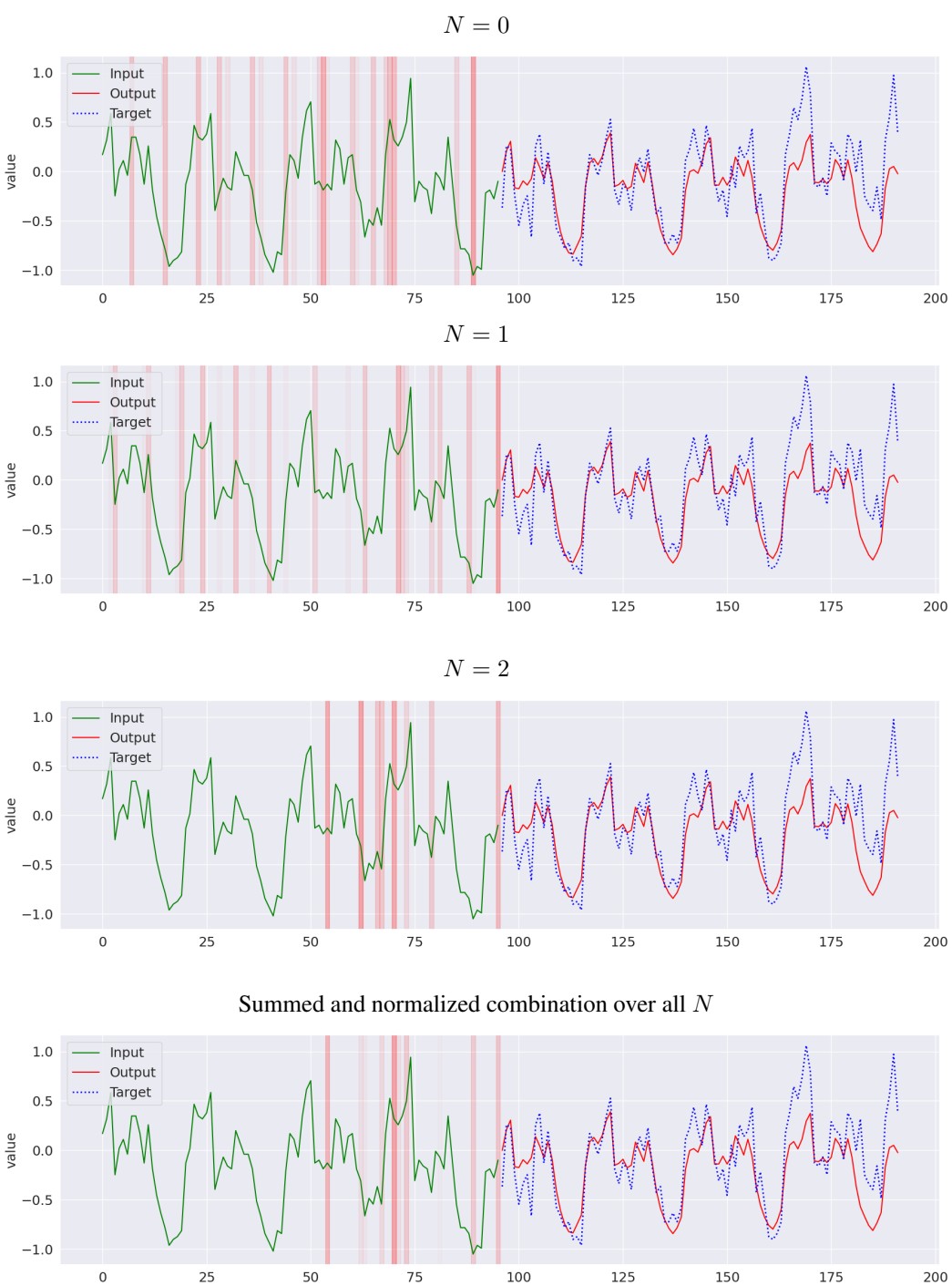

Figure 3: Highlighted attention weights during an ETTh1 forecasting task for all 3 *EncLayer*s, starting with the first *EncLayer* at the top and concluding with the the combined version over all *EncLayer* at the bottom.

Table 6: Best hyperparamter configurations for the TSRM model for the forecasting task.

| Dataset | Config | N | d | h | R (kernel_size, dilation, groups [-1=depthwise-convolution]) | attention |
|---|---|---|---|---|---|---|
| ELC | T=96, H=96 | 4 | 16 | 8 | [[3, 1, -1], [5, 2, -1], [10, 3, -1]] | entmax15 |
| ETTm1 | T=96, H=96 | 4 | 128 | 16 | [[3, 1, 1], [10, 2, 1], [15, 3, 1]] | entmax15 |
| ETTm2 | T=96, H=96 | 8 | 64 | 32 | [[3, 1, 1], [10, 2, 1], [15, 3, 1]] | entmax15 |
| ETTh1 | T=96, H=96 | 4 | 128 | 16 | [[3, 1, -1], [5, 2, -1], [10, 3, -1]] | entmax15 |
| ETTh2 | T=96, H=96 | 3 | 64 | 16 | [[3, 1, -1], [5, 2, -1], [10, 3, -1]] | entmax15 |
| Weather | T=96, H=96 | 4 | 128 | 32 | [[3, 1, 1]] | entmax15 |
| Exchange | T=96, H=96 | 4 | 64 | 16 | [[3, 1, 1], [5, 2, 1], [10, 3, 1]] | entmax15 |
| ELC | T=96, H=192 | 3 | 32 | 8 | [[3, 1, -1], [5, 2, -1], [10, 3, -1]] | entmax15 |
| ETTm1 | T=96, H=192 | 5 | 128 | 16 | [[3, 1, 1], [10, 2, 1], [15, 3, 1]] | entmax15 |
| ETTm2 | T=96, H=192 | 4 | 128 | 64 | [[3, 1, -1], [5, 2, -1], [10, 3, -1]] | entmax15 |
| ETTh1 | T=96, H=192 | 4 | 64 | 16 | [[3, 1, 1], [10, 2, 1], [15, 3, 1]] | entmax15 |
| ETTh2 | T=96, H=192 | 4 | 128 | 32 | [[3, 1, 1], [10, 2, 1], [15, 3, 1]] | entmax15 |
| Weather | T=96, H=192 | 2 | 128 | 4 | [[3, 1, 1], [5, 2, 1], [10, 3, 1]] | entmax15 |
| Exchange | T=96, H=192 | 4 | 64 | 4 | [[3, 1, 1], [5, 2, 1], [10, 3, 1]] | entmax15 |
| ELC | T=96, H=336 | 3 | 32 | 8 | [[3, 1, -1], [5, 2, -1], [10, 3, -1]] | entmax15 |
| ETTm1 | T=96, H=336 | 4 | 128 | 32 | [[3, 1, 1], [10, 2, 1], [15, 3, 1]] | entmax15 |
| ETTm2 | T=96, H=336 | 8 | 64 | 16 | [[3, 1, -1], [5, 2, -1], [10, 3, -1]] | entmax15 |
| ETTh1 | T=96, H=336 | 4 | 128 | 16 | [[3, 1, 1], [10, 2, 1], [15, 3, 1]] | entmax15 |
| ETTh2 | T=96, H=336 | 3 | 128 | 16 | [[3, 1, 1], [10, 2, 1], [15, 3, 1]] | entmax15 |
| Weather | T=96, H=336 | 4 | 128 | 4 | [[3, 1, 1], [5, 2, 1], [10, 3, 1]] | entmax15 |
| Exchange | T=96, H=336 | 2 | 64 | 32 | [[3, 1, -1], [5, 2, -1], [10, 3, -1]] | entmax15 |
| ELC | T=96, H=720 | 4 | 32 | 8 | [[3, 1, -1], [5, 2, -1], [10, 3, -1]] | entmax15 |
| ETTm1 | T=96, H=720 | 4 | 128 | 32 | [[3, 1, 1], [10, 2, 1], [15, 3, 1]] | entmax15 |
| ETTm2 | T=96, H=720 | 2 | 128 | 16 | [[3, 1, 1], [10, 2, 1], [15, 3, 1]] | entmax15 |
| ETTh1 | T=96, H=720 | 4 | 64 | 32 | [[3, 1, -1], [5, 2, -1], [10, 3, -1]] | entmax15 |
| ETTh2 | T=96, H=720 | 6 | 64 | 16 | [[3, 1, 1], [10, 2, 1], [15, 3, 1]] | entmax15 |
| Weather | T=96, H=720 | 2 | 64 | 8 | [[3, 1, 1], [5, 2, 1], [10, 3, 1]] | entmax15 |
| Exchange | T=96, H=720 | 2 | 128 | 4 | [[3, 1, 1], [5, 2, 1], [10, 3, 1]] | entmax15 |

Table 7: Mean and STD of MSE and MAE across 5 runs for the ETTh2 and Weather datasets in the format Mean ± STD.

| Dataset | MSE | MAE |
|---|---|---|
| ETTh2 | 0.289±0.028 | 0.336±0.018 |
| Weather | 0.162±0.013 | 0.204±0.016 |

Table 8: Performance comparison for the multivariate imputation task with missing ratios $r_m \in 0.125, 0.25, 0.375, 0.5$ and a fixed input length of 96. Bold/underline indicate best/second.

| Models | | TSRM | | TSRM_IFC | | TimesNet | | ETSformer | | LightTS | | DLinear | | FEDformer | | Stationary | | Autoformer | | Pyraformer | | Informer | | LogTrans | |
|---|---|---|---|---|---|---|---|---|---|---|---|---|---|---|---|---|---|---|---|---|---|---|---|---|---|
| | $r_m$ | MSE | MAE | MSE | MAE | MSE | MAE | MSE | MAE | MSE | MAE | MSE | MAE | MSE | MAE | MSE | MAE | MSE | MAE | MSE | MAE | MSE | MAE | MSE | MAE |
| ECL | 0.125 | **0.057** | **0.146** | 0.060 | 0.166 | 0.085 | 0.202 | 0.196 | 0.321 | 0.102 | 0.229 | 0.092 | 0.214 | 0.107 | 0.237 | 0.093 | 0.210 | 0.089 | 0.210 | 0.297 | 0.383 | 0.218 | 0.326 | 0.164 | 0.296 |
| | 0.250 | **0.065** | **0.166** | 0.067 | 0.168 | 0.089 | 0.206 | 0.207 | 0.332 | 0.121 | 0.252 | 0.120 | 0.247 | 0.120 | 0.251 | 0.097 | 0.214 | 0.096 | 0.220 | 0.294 | 0.380 | 0.219 | 0.326 | 0.169 | 0.299 |
| | 0.375 | **0.072** | **0.166** | 0.083 | 0.191 | 0.094 | 0.213 | 0.219 | 0.344 | 0.141 | 0.273 | 0.144 | 0.276 | 0.136 | 0.266 | 0.102 | 0.220 | 0.104 | 0.229 | 0.296 | 0.381 | 0.222 | 0.328 | 0.178 | 0.305 |
| | 0.50 | 0.098 | 0.203 | **0.082** | **0.191** | 0.100 | 0.221 | 0.235 | 0.357 | 0.160 | 0.293 | 0.175 | 0.305 | 0.158 | 0.284 | 0.108 | 0.228 | 0.113 | 0.239 | 0.299 | 0.383 | 0.228 | 0.331 | 0.187 | 0.312 |
| | AVG | **0.073** | **0.170** | 0.073 | 0.179 | 0.092 | 0.210 | 0.214 | 0.338 | 0.131 | 0.262 | 0.132 | 0.260 | 0.130 | 0.260 | 0.100 | 0.218 | 0.100 | 0.224 | 0.296 | 0.382 | 0.222 | 0.328 | 0.174 | 0.303 |
| ETTm1 | 0.125 | 0.039 | 0.123 | 0.033 | 0.119 | **0.019** | **0.092** | 0.067 | 0.188 | 0.075 | 0.180 | 0.058 | 0.162 | 0.035 | 0.135 | 0.026 | 0.107 | 0.034 | 0.124 | 0.670 | 0.541 | 0.047 | 0.155 | 0.041 | 0.141 |
| | 0.250 | 0.039 | 0.123 | 0.034 | 0.121 | **0.023** | **0.101** | 0.096 | 0.229 | 0.093 | 0.206 | 0.080 | 0.193 | 0.052 | 0.166 | 0.032 | 0.119 | 0.046 | 0.144 | 0.689 | 0.553 | 0.063 | 0.180 | 0.044 | 0.144 |
| | 0.375 | 0.043 | 0.130 | 0.046 | 0.139 | **0.029** | **0.111** | 0.133 | 0.271 | 0.113 | 0.231 | 0.103 | 0.219 | 0.069 | 0.191 | 0.039 | 0.131 | 0.057 | 0.161 | 0.737 | 0.581 | 0.079 | 0.200 | 0.052 | 0.158 |
| | 0.50 | 0.051 | 0.142 | 0.072 | 0.185 | **0.036** | **0.124** | 0.186 | 0.323 | 0.134 | 0.255 | 0.132 | 0.248 | 0.089 | 0.218 | 0.047 | 0.145 | 0.067 | 0.174 | 0.770 | 0.605 | 0.093 | 0.218 | 0.063 | 0.173 |
| | AVG | 0.043 | 0.130 | 0.046 | 0.141 | **0.027** | **0.107** | 0.120 | 0.253 | 0.104 | 0.218 | 0.093 | 0.206 | 0.061 | 0.178 | 0.036 | 0.126 | 0.051 | 0.151 | 0.716 | 0.570 | 0.070 | 0.188 | 0.050 | 0.154 |
| ETTm2 | 0.125 | 0.025 | 0.091 | 0.020 | 0.087 | **0.018** | **0.080** | 0.108 | 0.239 | 0.034 | 0.127 | 0.062 | 0.166 | 0.056 | 0.159 | 0.021 | 0.088 | 0.023 | 0.092 | 0.394 | 0.470 | 0.133 | 0.270 | 0.103 | 0.229 |
| | 0.250 | 0.027 | 0.095 | 0.021 | 0.096 | **0.020** | **0.085** | 0.164 | 0.294 | 0.042 | 0.143 | 0.085 | 0.196 | 0.080 | 0.195 | 0.024 | 0.096 | 0.026 | 0.101 | 0.421 | 0.482 | 0.135 | 0.272 | 0.120 | 0.248 |
| | 0.375 | 0.029 | 0.109 | 0.024 | 0.102 | **0.023** | **0.091** | 0.237 | 0.356 | 0.051 | 0.159 | 0.106 | 0.222 | 0.110 | 0.231 | 0.027 | 0.103 | 0.030 | 0.108 | 0.478 | 0.521 | 0.155 | 0.293 | 0.138 | 0.260 |
| | 0.50 | 0.033 | 0.118 | 0.037 | 0.127 | **0.026** | **0.098** | 0.323 | 0.421 | 0.059 | 0.174 | 0.131 | 0.247 | 0.156 | 0.276 | 0.030 | 0.108 | 0.035 | 0.119 | 0.568 | 0.560 | 0.200 | 0.333 | 0.117 | 0.247 |
| | AVG | 0.028 | 0.103 | 0.026 | 0.103 | **0.022** | **0.088** | 0.208 | 0.328 | 0.046 | 0.151 | 0.096 | 0.208 | 0.100 | 0.215 | 0.026 | 0.099 | 0.028 | 0.105 | 0.465 | 0.508 | 0.156 | 0.292 | 0.120 | 0.246 |
| ETTh1 | 0.125 | 0.091 | 0.199 | **0.046** | **0.146** | 0.057 | 0.159 | 0.126 | 0.263 | 0.240 | 0.345 | 0.151 | 0.267 | 0.070 | 0.190 | 0.060 | 0.165 | 0.074 | 0.182 | 0.857 | 0.609 | 0.114 | 0.234 | 0.229 | 0.330 |
| | 0.250 | 0.104 | 0.212 | **0.052** | **0.158** | 0.069 | 0.178 | 0.169 | 0.304 | 0.265 | 0.364 | 0.180 | 0.292 | 0.106 | 0.236 | 0.080 | 0.189 | 0.090 | 0.203 | 0.829 | 0.672 | 0.140 | 0.262 | 0.207 | 0.323 |
| | 0.375 | 0.112 | 0.221 | **0.066** | **0.173** | 0.084 | 0.196 | 0.220 | 0.347 | 0.296 | 0.382 | 0.215 | 0.318 | 0.124 | 0.258 | 0.102 | 0.212 | 0.109 | 0.222 | 0.830 | 0.675 | 0.174 | 0.293 | 0.210 | 0.328 |
| | 0.50 | 0.117 | 0.225 | 0.181 | 0.245 | **0.102** | **0.215** | 0.293 | 0.402 | 0.334 | 0.404 | 0.257 | 0.347 | 0.165 | 0.299 | 0.133 | 0.240 | 0.137 | 0.248 | 0.854 | 0.691 | 0.215 | 0.325 | 0.230 | 0.348 |
| | AVG | 0.106 | 0.214 | 0.086 | 0.180 | **0.078** | **0.187** | 0.202 | 0.329 | 0.284 | 0.374 | 0.201 | 0.306 | 0.116 | 0.246 | 0.094 | 0.202 | 0.102 | 0.214 | 0.842 | 0.662 | 0.161 | 0.278 | 0.219 | 0.332 |
| ETTh2 | 0.125 | 0.060 | 0.156 | 0.045 | 0.141 | **0.040** | **0.130** | 0.187 | 0.319 | 0.101 | 0.231 | 0.100 | 0.216 | 0.095 | 0.212 | 0.042 | 0.133 | 0.044 | 0.138 | 0.976 | 0.754 | 0.305 | 0.431 | 0.173 | 0.308 |
| | 0.250 | 0.068 | 0.166 | 0.046 | 0.149 | **0.046** | **0.141** | 0.279 | 0.390 | 0.115 | 0.246 | 0.127 | 0.247 | 0.137 | 0.258 | 0.049 | 0.147 | 0.050 | 0.149 | 1.037 | 0.774 | 0.322 | 0.444 | 0.175 | 0.310 |
| | 0.375 | 0.069 | 0.168 | 0.067 | 0.170 | **0.052** | **0.151** | 0.400 | 0.465 | 0.126 | 0.257 | 0.158 | 0.276 | 0.187 | 0.304 | 0.056 | 0.158 | 0.060 | 0.163 | 1.107 | 0.800 | 0.353 | 0.462 | 0.185 | 0.315 |
| | 0.50 | 0.165 | 0.261 | 0.073 | 0.178 | **0.060** | **0.162** | 0.602 | 0.572 | 0.136 | 0.268 | 0.183 | 0.299 | 0.232 | 0.341 | 0.065 | 0.170 | 0.068 | 0.173 | 1.193 | 0.838 | 0.369 | 0.472 | 0.212 | 0.339 |
| | AVG | 0.090 | 0.188 | 0.058 | 0.160 | **0.050** | **0.146** | 0.367 | 0.436 | 0.120 | 0.250 | 0.142 | 0.260 | 0.163 | 0.279 | 0.053 | 0.152 | 0.056 | 0.156 | 1.078 | 0.792 | 0.337 | 0.452 | 0.186 | 0.318 |
| Weather | 0.125 | 0.025 | 0.044 | 0.024 | 0.043 | 0.025 | 0.045 | 0.057 | 0.141 | 0.047 | 0.101 | 0.039 | 0.084 | 0.041 | 0.107 | 0.027 | 0.051 | 0.026 | 0.047 | 0.140 | 0.220 | 0.037 | 0.093 | 0.037 | 0.072 |
| | 0.250 | 0.029 | 0.046 | 0.026 | 0.040 | 0.029 | 0.046 | 0.065 | 0.155 | 0.052 | 0.111 | 0.048 | 0.103 | 0.064 | 0.163 | 0.029 | 0.056 | 0.030 | 0.054 | 0.147 | 0.229 | 0.042 | 0.100 | 0.038 | 0.074 |
| | 0.375 | 0.031 | 0.049 | 0.029 | 0.047 | 0.031 | 0.047 | 0.081 | 0.180 | 0.058 | 0.121 | 0.057 | 0.117 | 0.107 | 0.229 | 0.033 | 0.062 | 0.032 | 0.060 | 0.156 | 0.240 | 0.049 | 0.111 | 0.039 | 0.078 |
| | 0.50 | 0.038 | 0.054 | 0.037 | 0.051 | 0.034 | 0.062 | 0.102 | 0.207 | 0.065 | 0.133 | 0.066 | 0.134 | 0.183 | 0.312 | 0.037 | 0.068 | 0.037 | 0.067 | 0.164 | 0.249 | 0.053 | 0.114 | 0.042 | 0.082 |
| | AVG | 0.031 | 0.048 | **0.029** | **0.045** | 0.030 | 0.054 | 0.076 | 0.171 | 0.056 | 0.116 | 0.052 | 0.110 | 0.099 | 0.203 | 0.032 | 0.059 | 0.031 | 0.057 | 0.152 | 0.234 | 0.045 | 0.104 | 0.039 | 0.076 |

Table 9: Hyperparameter sensitivity study for $N$.

| Dataset | N=0 | | N=1 | | N=2 | | N=3 | | N=4 | | N=5 | | N=6 | | N=7 | | N=8 | |
|---|---|---|---|---|---|---|---|---|---|---|---|---|---|---|---|---|---|---|
| | MSE | MAE | MSE | MAE | MSE | MAE | MSE | MAE | MSE | MAE | MSE | MAE | MSE | MAE | MSE | MAE | MSE | MAE |
| ETTh1 | 0.406 | 0.411 | 0.381 | 0.395 | 0.393 | 0.401 | 0.383 | **0.394** | 0.398 | 0.403 | 0.398 | 0.403 | 0.384 | 0.397 | **0.379** | **0.394** | 0.396 | 0.407 |
| Weather | 0.217 | 0.248 | 0.163 | 0.205 | 0.162 | **0.204** | 0.162 | **0.204** | **0.161** | **0.204** | 0.162 | 0.204 | 0.162 | 0.205 | 0.167 | 0.298 | 0.168 | 0.210 |

- Number of *EncLayer*s: $N \in [0, 1, \dots 12]$
- Number of heads in the self-attention module: $h \in \{2, 4, 8, 16, 32\}$
- Feature embedding size: $d \in \{8, 16, 32, 64, 128\}$
- Attention function: $attention\_func \in \{$classic (vanilla) Vaswani et al. (2017), sparse-attention (entmax15) Wu et al. (2020b)$\}$.
- Amount and configuration of the CNN layers in the *ReprLayer*: We varied the amount of CNN layers between 1 and 4. The configuration was designed such that the smallest kernel covered around 3 values and the biggest around 50% - 80% of the input sequence. The kernels in between covered middle sized sequences. We also experimented with different amounts of groups in the CNN layers, as well as with depthwise-convolution to further decrease the memory footprint.

## A.5 IMPUTATION

We extend our evaluation from Section 4.3 with a more detailed evaluation, including more SOTA approaches for the comparison, and we provide the results of all missing ratios ($r_m$) separately. Please see Table 8. For the evaluation, we consider six datasets(i.e., ECL, ETTm1, ETTm2, ETTh1, ETTh2, Weather) and compare against multiple SOTA techniques: TimesNet (Wu et al., 2022), ETSformer (Woo et al., 2022b), LightTS (Zhang et al., 2022a), DLinear (Zeng et al., 2022), FEDformer Zhou et al. (2022), Stationary (Liu et al., 2022b), Autoformer (Wu et al., 2021), Pyraformer (Liu et al., 2021), Informer (Zhou et al., 2021), and LogTrans Li et al. (2019).

## A.6 ABLATION STUDY

In addition to the ablation study results in Section 5, we provide more detailed results for each experiment. Table 9 shows all details, including MSE and MAE metrics for both datasets, of the sensitivity study for the parameter $N$. We further provide the exact results of the architecture variation study in Table 10. Here, we provide the MSE and MAE metrics for both datasets, Weather and ETTh1.

Table 10: Ablation study with different architectural variations.

| Dataset | TSRM | | TSRM_IFC | | no_merge | | no_merge+$R1$ | | no_merge+$R0$ | |
|---|---|---|---|---|---|---|---|---|---|---|
| | MSE | MAE | MSE | MAE | MSE | MAE | MSE | MAE | MSE | MAE |
| ETTh1 | **0.379** | **0.394** | 0.387 | 0.401 | 0.382 | **0.394** | 0.398 | 0.403 | 0.405 | 0.416 |
| Weather | 0.161 | 0.202 | **0.153** | **0.200** | 0.165 | 0.206 | 0.165 | 0.206 | 0.166 | 0.206 |

