# OpenReview forum: "Time Series Representation Models for Multivariate Time Series Forecasting and Imputation"
_ICLR.cc/2025/Conference — Submitted to ICLR 2025_

### Official Review · Reviewer_Khpu · 2024-10-27

**Soundness:** 2
**Presentation:** 2
**Contribution:** 2
**Rating:** 5
**Confidence:** 4

**Summary:**

This paper proposes a multilayered representation learning architecture for multivariate time series forecasting and imputation, achieving superior performance on seven public datasets.

**Strengths:**

The methodology presentation is clear, the code is provided, and the performance surpasses multiple models.

**Weaknesses:**

1. **Unclear motivation:** The paper does not discuss which limitations of existing methods it aims to address, lacking a direct explanation in this regard.
2. **Limited scope of tasks:** Although the paper proposes a representation learning approach, it is only applied to forecasting and imputation, which are tasks of the same nature. I hope to see applications in classification and anomaly detection as well.
3. **Lack of discussion and discussion on most related works:** The paper compares only with end-to-end models, but it misses comparisons with the most relevant representation-based methods, such as Cost[1]. Additionally, recent foundation models, such as MOMENT[3] and Moirai[2], could also be leveraged as representations. Please discuss differences and make comparisions.

[1] Woo, Gerald, et al. "CoST: Contrastive Learning of Disentangled Seasonal-Trend Representations for Time Series Forecasting." International Conference on Learning Representations.


[2] Woo, Gerald, et al. "Unified Training of Universal Time Series Forecasting Transformers." Forty-first International Conference on Machine Learning.


[3] Goswami, Mononito, et al. "MOMENT: A Family of Open Time-series Foundation Models." Forty-first International Conference on Machine Learning.

**Questions:**

Please check limitations.

---

> ### Author Response · Authors · 2024-11-18
>
> Thank you for your review of our paper. We will go into the individual points in more detail below:
>
> > ***Unclear motivation:** The paper does not discuss which limitations of existing methods it aims to address, lacking a direct explanation in this regard.*
>
> You are right, this point is somewhat lost in the current motivation. We have slightly extended the motivation with:
> *In this paper, we extend this concept [of TimesNet i.a.] of learning temporal representations by introducing a lightweight and adaptive multidimensional framework with a hierarchical design and high configurability to handle complex temporal variations and be applicable to many datasets.*
>
> > ***Limited scope of tasks:** Although the paper proposes a representation learning approach, it is only applied to forecasting and imputation, which are tasks of the same nature. I hope to see applications in classification and anomaly detection as well.*
>
> The term “representation learning” in this work refers to the learning and extraction of temporal features from the input time series. We are aware that there are other definitions of representation learning that go in the direction of pretraining and finetuning, such as PatchTST, which is then also used for a wider range of tasks. However, the initial focus of this work was to present and evaluate the architecture in the context of forecasting and imputation. Future work will address classification, anomaly detection, and clustering to the same extent, and preliminary experiments in this direction are already ongoing. We have included these points as an outlook in the conclusion of the manuscript.
>
> > ***Lack of discussion and discussion on most related works:** The paper compares only with end-to-end models, but it misses comparisons with the most relevant representation-based methods, such as Cost[1]. Additionally, recent foundation models, such as MOMENT[3] and Moirai[2], could also be leveraged as representations. Please discuss differences and make comparisions.*
>
> Thank you for bringing this work to our attention. We have reviewed it and added it to our related work. However, a comparison with our approach is not intended as the linked approaches, although they present interesting concepts, are either not comparable in terms of the reported performance [1], or follow a different concept, such as pre-training/foundation models [2,3], which is not comparable to ours as we follow a pure downstream approach. However, in future work, we will also evaluate our proposed architecture in the context of foundation models and include the aforementioned approaches in our performance comparison.

---

> ### Author Response · Authors · 2024-11-23
>
> We would like to thank you again for the review and would like to politely ask if there are any further questions or comments that we can help with.

---

### Official Review · Reviewer_hGyz · 2024-11-02

**Soundness:** 2
**Presentation:** 2
**Contribution:** 2
**Rating:** 3
**Confidence:** 4

**Summary:**

This paper introduces new architecture for forecasting/imputation of time-series. Overall, it introduces encoder blocks comprising of a Representation layer and Merge layer with self-attention in between. The representation and merge layers have no non-linearities and consist of convolution/transposed-convolution layers respectively. The models are evaluated on forecasting/imputation tasks.

**Strengths:**

* Clear presentation and informative figure 1.
* Experiments on both forecasting/imputation.
* TSRM are lighter than popular models such as TimesNet/Patch-TST in terms of parameter-counts.

**Weaknesses:**

* The paper introduces a lot of hyperparameters. Based on Appendix A.4, the paper searches over 13 possible EncLayers $\times$ 5 possible attention-heads $\times$ 5 possible feature-embedding sizes $\times$ 2 possible attention-functions $\times$ (at least) 4 possible CNN configurations = 2600 configurations!
* From the table having best hyperparameters, sparse-attention seems to be the best combination across all datasets. But, there seems to be significant variation across different datasets/horizons. Even for the same dataset, different horizon seems to have a different optimal configuration.
* Although TSRM is lighter than other models and this may help accelerate the search, the computational cost of picking the right hyperparameters cannot be ignored. Also, I couldn't find any analysis of wallclock times (train/inference) in the paper.
* Table 7 reports the Mean/Std across 5 runs of ETTh2 and weather. From my understanding, this mean/std is computed for running the model with optimal configuration for 5 times. In practice, the hyperparameters should be selected based on 5 runs and then, the corresponding test-performance should be reported. However, this may be very expensive given the hyperparameter space.
* TimesNet seems to be uniformly better than TSRM on the imputation task. Also, PatchTST is not considered for the imputation task.
* The model is trained on forecasting/imputation tasks only. Self-supervised learning may also be explored to establish the generality of this architecture. However, the hyperparameter-search may make this expensive.
* Given the huge hyperparameter space of TSRM, it may be fairer to allow PatchTST to select its patch-size/input-length. Under optimal Patch-TST performance, does TSRM still perform favorably?

**Questions:**

Please see weaknesses.
1) What is $\alpha$ in Eq 4?
2) How does TSRM generalize to different missing ratios at test time?

---

> ### Author Response · Authors · 2024-11-18
>
> Thank you for your review of our paper. We will go into the individual points in more detail below:
>
> > *The paper introduces a lot of hyperparameters. Based on Appendix A.4, the paper searches over 13 possible EncLayers $\times$ 5 possible attention-heads $\times$ 5 possible feature-embedding sizes $\times$ 2 possible attention-functions (at least) 4 possible CNN configurations = 2600 configurations!*
>
> You are right that our architecture involves a large number of possible hyperparameters. However, in practice, fewer would be used, as our analyses have shown that sparse-attention, for example, is superior to vanilla attenion in our architecture throughout our experiments. From the outset, however, this realization is not self-evident, which is why we originally had to investigate it and therefore introduced it as a hyperparameter. The same applies to some other hyperparameters, which we did not simply assume to be ‘magic numbers’ but for which we wanted to find empirically meaningful values. Therefore, we originally had to include them in the experiments to assess their impact on performance.
>
> However, the results show that some hyperparameters work best in a certain range. In future applications of our approach, they therefore no longer need to be investigated to the same extent in further experiments.
>
> Nevertheless, we would like to point out that these findings could not be assumed from the beginning and were therefore an important part of our contribution.
>
> However, an error crept into the manuscript, which referred to a systematic grid search. In fact, we worked with a random search, trastically reducing the above mentioned number of investigated configurations of 2600. We have corrected the error in the manuscript.
>
> > *From the table having best hyperparameters, sparse-attention seems to be the best combination across all datasets. But, there seems to be significant variation across different datasets/horizons. Even for the same dataset, different horizon seems to have a different optimal configuration.*
>
> We have added Table 6 for the sake of completeness and to promote the reproducibility of our reported results. However, it only shows the best configurations of a forecasting configuration of a dataset. This does not necessarily mean that the same configuration performed significantly worse for different horizons.
> Even if your conclusion based on Table 6 is understandable, we would like to point out that in practice, due to the contents of the table with only the very best setting in each case, this cannot be derived from the table alone, as it does not fully represent all settings.
>
> > *Although TSRM is lighter than other models and this may help accelerate the search, the computational cost of picking the right hyperparameters cannot be ignored.*
>
> As explained above, the hyperparameters listed were of great importance for the introduction and development of the approach, but will be significantly reduced in further experiments or applications (given that our extensive parameter study has already found a certain range of well-functioning values for many hyperparameters).
>
> > *Also, I couldn't find any analysis of wallclock times (train/inference) in the paper.*
>
> We have not specified these values as they are heavily dependent on the underlying hardware and other processes currently running on the machine, especially on the GPU. In our opinion, they are therefore hardly representative. Nevertheless, in order to give an impression of the runtime, we have carried out a runtime analysis for the ETTh1 and the Weather dataset and added it to the manuscript in the Appendix (A.3).
>
> > *Table 7 reports the Mean/Std across 5 runs of ETTh2 and weather. From my understanding, this mean/std is computed for running the model with optimal configuration for 5 times. In practice, the hyperparameters should be selected based on 5 runs and then, the corresponding test-performance should be reported. However, this may be very expensive given the hyperparameter space.*
>
> Similar to prior work, for instance, Informer by Zhou et al., we found the best hyperparameters based on the test performance. Then, using the best hyperparameters found, we repeated training and testing for multiple different random seeds. This allows us to assess the robustness and reliability of experimental results, as required in many AI conferences, such as NeurIPS. Table 7 shows that the results are stable across different random seeds.

---

> ### Author Response · Authors · 2024-11-18
>
> > *TimesNet seems to be uniformly better than TSRM on the imputation task.*
>
> You are right that TimesNet is better at some metrics and datasets. This was discussed in our results section, pointing out favourable performance of TSRM_IFC on ECL and Weather, while TimesNet prevails on the ETT variants.
>
> > *Also, PatchTST is not considered for the imputation task.*
>
> PatchTST was introduced primarily as a forecasting model, not as an imputation model. To the best of our knowledge, PatchTST imputation has never been evaluated in a scientific setting and is not listed as a comparison in TimesNet, from which we adopted the experimental setup [1]. Accordingly, we have not used PatchTST as a comparison.
>
> > *The model is trained on forecasting/imputation tasks only. Self-supervised learning may also be explored to establish the generality of this architecture. However, the hyperparameter-search may make this expensive.*
>
> This is a good idea, and we are already planning to consider self-supervised learning in future work. We are also planning further experiments with additional tasks and experiments in the direction of foundation models, where experiments regarding classification are, for example, already ongoing.
> We have added this to the conclusion. As already mentioned above, we are confident that this paper's extensive parameter study allows for a significant reduction of the hyperparameter search space for future experiments.
>
> > *Given the huge hyperparameter space of TSRM, it may be fairer to allow PatchTST to select its patch-size/input-length. Under optimal Patch-TST performance, does TSRM still perform favorably?*
>
> We agree with the argument that PatchTST should be able to choose the optimal patch size. However, PatchTST has already been trained with the best configuration, as reported by Wu et al. [1]. We have adopted these values.
> The choice of input length is independent of the hyperparameters. We understand that this point can be controversial. However, we argue that comparability is better when all models are trained with the same input size. Moreover, using the same input length corresponds to the procedure adopted by most of the approaches we compared against (TimesNet, iTransformer, Crossformer, ...), which is why we also stuck to it in our approach. Furthermore, the training effort for all architectures, not only ours, would be enormous otherwise, since all of them would have to be trained with four input lengths on four horizon lengths, and this with all hyperparameters individually.
>
>
> > *What is $\alpha$ in Eq 4?*
>
> It is a weighting factor, set to the inverse of the missing ratio, to reconcile the loss of masked and unmasked values. For clarity, we replaced $\alpha$ by $\frac{1}{r_m}$ and added an explanation to the manuscript.
>
> > *How does TSRM generalize to different missing ratios at test time?*
>
> We have not evaluated this, but we are including this aspect in future work.
>
> [1] Wu, H., Hu, T., Liu, Y., Zhou, H., Wang, J., & Long, M. (2022). Timesnet: Temporal 2d-variation modeling for general time series analysis. arXiv preprint arXiv:2210.02186.

---

> > ### Comment · Reviewer_hGyz · 2024-11-18
> > **Thank You!**
> >
> > I thank the authors for the rebuttal response. Directly comparing the results included in the paper with the official results of PatchTST, I find that this method does not perform competitively with Patch-TST on forecasting. Even with extensive hyperparameter tuning of TSRM, it sometimes falls far below the official Patch-TST performance and I am hoping that the authors can explain when one should use TSRM instead of Patch-TST (for forecasting/self-supervised tasks) or TimesNet (for imputation).

---

> ### Author Response · Authors · 2024-11-18
> **TSRM vs. PatchTST/TimesNet**
>
> Thank you for the follow-up question, which we are glad to address. It is accurate that the official results reported for PatchTST surpass those presented in our work. However, it is important to note a key difference in the experimental setup: the results for PatchTST, as well as for the comparison models in its paper, are obtained by treating the input length as a tunable hyperparameter. This approach allows PatchTST to optimize its performance based on the input length, making these results fundamentally incomparable to our findings, where all architectures were trained with a fixed input length of 96.
>
> Additionally, the strong performance of PatchTST is often linked to significantly long input lengths. For instance, in the case of the Weather dataset, the results are derived using an input length of 720 (as documented in Table 9 of Nie et al. [1]), which incurs substantial computational overhead. These computational costs are further amplified during hyperparameter optimization, as the process necessitates tuning for every combination of input length and forecasting horizon.
>
> A direct comparison between the two architectures under variable input lengths is not feasible, making it inappropriate to declare one architecture universally "better" in that context. However, by evaluating both architectures under the constraint of a fixed input length, we provide a clear basis for assessing their relative performance within this controlled scenario.
>
> Moreover, our proposed architecture demonstrates significantly greater parameter efficiency across all tested datasets, with an average of 0.904 million trainable parameters compared to 6.903 million for PatchTST (see Table 3). This leaner architecture highlights an advantage in computational efficiency, which is particularly relevant for resource-constrained applications.
>
> Nevertheless it is important to mention that PatchTST has pioneered patch-based TS analysis and has contributed significantly to the further development of Transformer-based TS analysis.
>
> In response to the question regarding TimesNet, our findings indicate no definitive recommendation based solely on performance, as results vary depending on the dataset. Specifically, TimesNet demonstrates superior performance on the ETTm1, ETTm2, and ETTh2 datasets, whereas our proposed TSRM architecture outperforms on the ECL, ETTh1, and Weather datasets (see Table 8). Consequently, the choice between these architectures is highly dependent on the specific dataset characteristics and requirements.
>
> However, our TSRM architecture offers a considerable advantage in terms of parameter efficiency, with substantially fewer trainable parameters on average compared to TimesNet.
>
> [1] Nie, Y., Nguyen, N. H., Sinthong, P., & Kalagnanam, J. (2022). A time series is worth 64 words: Long-term forecasting with transformers. arXiv preprint arXiv:2211.14730.

---

> > ### Comment · Reviewer_hGyz · 2024-11-18
> > **Thank you!**
> >
> > I thank the authors for kindly sharing their perspective about TSRM vs PatchTST/TimesNet! In summary, the argument seems to be that:
> > 1. Not fair to compare PatchTST that uses a lookback length of 336 with TSRM that uses a lookback length of 96.
> > 2. The trainable parameters are fewer in the case of TSRM vs PatchTST and this should likely help with compute-efficiency.
> > 3. The difference between TimesNet and PatchTST is domain-dependent.
> >
> > In my understanding, lookback-length is another hyperparameter of the model. Could you please share how many hyperparameters of the Patch-TST model were optimized and compare it with the number of hyperparameters optimized for TSRM? On the other hand, a natural curiosity is "does TSRM continue to outperform Patch-TST at longer lookback windows"? Also, see [1] which considers a fair benchmark by allowing each method to independently select the best lookback length.
> >
> > Regarding the trainable parameters and compute-efficiency: there is no empirical evidence that TSRM is more compute-efficient than Patch-TST. While the trainable parameters of TSRM are fewer, it is surprising to see that Patch-TST's standard deviations are an order of magnitude smaller (Table 14). This may also be a side-effect of the longer lookback window as alluded to in the response above but requires empirical verification.
> >
> > It may be helpful to include results on other domains (i.e., datasets) where TSRM outperforms TimesNet on the imputation task to make a stronger case that the weaker performance of TimesNet is only on ETT-datasets.
> >
> > Overall, I still have reservations about the empirical performance of this method given the extensive hyperparameter tuning (grid-search/random) and weaker baseline selection.
> >
> > [1] TFB: Towards Comprehensive and Fair Benchmarking of Time Series Forecasting Methods. (PVLDB 2024)
> >
> > Thank you again for engaging in a discussion with me!

---

> > > ### Author Response · Authors · 2024-11-19
> > >
> > > We thank you for the in-depth follow-up questions and will answer them individually below:
> > >
> > > > *1. Not fair to compare PatchTST that uses a lookback length of 336 with TSRM that uses a lookback length of 96.*
> > >
> > > It is not a matter of fair or unfair, but simply a different experimental setup that cannot be compared.
> > >
> > > > *2. The trainable parameters are fewer in the case of TSRM vs PatchTST and this should likely help with compute-efficiency.*
> > >
> > > An architecture that achieves comparable performance with current SOTA approaches, but requires considerably fewer trainable parameters is generally to be favored, or would you disagree here?
> > >
> > > > *3. The difference between TimesNet and PatchTST is domain-dependent.*
> > >
> > > We are not sure how you mean this or how this was to be inferred from our answer. Could you please provide some more context here? We did not write anything about a comparison between PatchTST and TimesNet in our previous answer.
> > >
> > > > *In my understanding, lookback-length is another hyperparameter of the model.*
> > >
> > > This is the case with PatchTST's chosen experimental setup, but not with ours.
> > >
> > > > *Could you please share how many hyperparameters of the Patch-TST model were optimized and compare it with the number of hyperparameters optimized for TSRM?*
> > >
> > > Unfortunately, the paper by PatchTST provides little information about the hyperparameters used, as there is no detailed list. However, the following hyperparameters can be identified from the code provided:
> > >
> > > 1. Amount of (Transformer) encoding layers
> > > 2. Attention heads
> > > 3. Feature embedding size
> > > 4. Pos. Feedforward size
> > > 5. Dropout
> > > 6. Head Dropout
> > > 7. Patch length
> > > 8. Patch stride
> > > 9. (input length)
> > >
> > > Our proposed TSRM architecture was optimized with the following hyperparameters:
> > >
> > > 1. N (Amount EncodingLayers)
> > > 2. Heads
> > > 3. Feature embedding size
> > > 4. Attention function
> > > 5. Representation Layer setup
> > >
> > > However, it is important to classify which hyperparameters were varied as part of a hyperparameter study and which were set. Only the hyperparameters that were varied appear in our list.
> > >
> > > As mentioned at the end of Section 4.2, the reported results of the Patch-TST model were taken from Liu et al. [1], who state that "all the compared baseline models that [they] reproduced are implemented based on the benchmark of TimesNet (Wu et al., 2023) Repository, which is fairly built on the configurations provided by each model’s original paper or official code." (Appendix A.2)
> > > According to Nie et al. [2], six sets of PatchTST hyperparameters were examined in Section A.6.2.
> > >
> > >
> > > > *On the other hand, a natural curiosity is "does TSRM continue to outperform Patch-TST at longer lookback windows"?*
> > >
> > > We share this curiosity and will investigate this in future work.
> > >
> > >
> > > > *Also, see [1] which considers a fair benchmark by allowing each method to independently select the best lookback length.*
> > >
> > > Thank you for pointing out this very interesting work. The paper describes two comparison methods, “fixed” and “rolling”. "fixed" is similar to our experimental setup, "rolling" is like that of PatchTST, but not exactly the same, since only an average of all lookback window lenghts are calculated, whereas PatchTST calculates the metrics with the same lookback window.
> > >
> > > Both methods are counted as part of the benchmark, but we cannot find a qualitative classification of the authors that would support your statement regarding the quality of one or the other method. Could you please provide a more precise indication of where this evaluation takes place in the paper?
> > >
> > >
> > > > *Regarding the trainable parameters and compute-efficiency: there is no empirical evidence that TSRM is more compute-efficient than Patch-TST.*
> > >
> > > We wrote in our previous answer “This leaner architecture highlights an advantage in computational efficiency [...]”. Would you disagree with this?
> > > However, we would argue that a shorter input sequence requires less VRAM than a long one and thus leads to better compute efficiency. Or would you disagree?
> > >
> > > [1] Liu, Y., Hu, T., Zhang, H., Wu, H., Wang, S., Ma, L., & Long, M. (2023). itransformer: Inverted transformers are effective for time series forecasting. arXiv preprint arXiv:2310.06625.
> > >
> > > [2] Nie, Y., Nguyen, N. H., Sinthong, P., & Kalagnanam, J. (2022). A time series is worth 64 words: Long-term forecasting with transformers. arXiv preprint arXiv:2211.14730.

---

> > > > ### Comment · Reviewer_hGyz · 2024-11-21
> > > > **Thank you!**
> > > >
> > > > Thank you for sharing your views about my concerns!
> > > >
> > > > 1. For the 9 hyperparameters you have listed, the official Patch-TST results seems to use identical hyperparameters across all datasets except (ILI, ETTh1, ETTh2). Of the 2600 configurations, how many configurations were tested in the random hyperparameter search?
> > > > 2. Please see section 5.1.2 of the TFB paper for their experimental methodology on how the lookback-lengths were selected. In my view, each method should be allowed to choose its optimal lookback-length, and it is indeed the policy followed in the TFB benchmark aiming to provide a fair benchmark of time-series forecasting methods.
> > > > 3. I disagree that experiments related to "does TSRM continue to outperform Patch-TST at longer lookback windows" constitutes future work. This is within the scope of the current submission, and it is important to provide a complete assessment of the strengths and weaknesses of a new method/architecture. In fact, TSRM may outperform PatchTST at longer lookback-windows as well leading to a new state-of-the-art across all settings!
> > > > 4. Having fewer parameters does not necessarily imply faster runtime: for e.g., depending on implementation, RNNs with fewer parameters may have longer inference times than Transformers with many more parameters. I agree with your earlier point that running-times are implementation dependent. However, a claim of faster-runtime/compute-efficiency needs to be empirically supported.
> > > >
> > > > ***
> > > > PS: Apologies for the typo in "The difference between TimesNet and PatchTST is domain-dependent." I meant "The difference between TimesNet and TSRM is domain-dependent."

---

> > > > > ### Author Response · Authors · 2024-11-23
> > > > >
> > > > > Thank you for the follow-up question.
> > > > >
> > > > > > *1. For the 9 hyperparameters you have listed, the official Patch-TST results seems to use identical hyperparameters across all datasets except (ILI, ETTh1, ETTh2). Of the 2600 configurations, how many configurations were tested in the random hyperparameter search?*
> > > > >
> > > > > During the experimentation phase, a certain insensitivity is usually observed for individual hyperparameters, which in our setup leads to a broader spectrum of hyperparameters, while other hyperparameters play a subordinate role, resulting in a reduction of the hyperparameter space. In our hyperparameter studies, we analyzed approximately 150 configurations per data set.
> > > > >
> > > > > > *2. Please see section 5.1.2 of the TFB paper for their experimental methodology on how the lookback-lengths were selected. In my view, each method should be allowed to choose its optimal lookback-length, and it is indeed the policy followed in the TFB benchmark aiming to provide a fair benchmark of time-series forecasting methods.*
> > > > > > *3. I disagree that experiments related to "does TSRM continue to outperform Patch-TST at longer lookback windows" constitutes future work. This is within the scope of the current submission, and it is important to provide a complete assessment of the strengths and weaknesses of a new method/architecture. In fact, TSRM may outperform PatchTST at longer lookback-windows as well leading to a new state-of-the-art across all settings!*
> > > > >
> > > > > We can follow the reasoning in the linked TFB paper, which proposes the rolling approach for multivariate forecasting. However, the proposed rolling approach does not correspond to the experimental setup of the PatchTST paper. In the rolling approach, the mean value of all lookback variants would be calculated.
> > > > >
> > > > > We can also understand the argumentation that some authors prefer the variant with variable lookback windwows. However, for the reasons explained above, we have opted for a fixed lookback window and set up our experimental setup accordingly. We follow the experimental setup of previous papers such as TimesNet, iTransformer, and FEDformer.
> > > > >
> > > > > Furthermore, in the time frame of this discussion phase, an empirically clean analysis of different lookback windows is not feasible.
> > > > >
> > > > > > *4. Having fewer parameters does not necessarily imply faster runtime: for e.g., depending on implementation, RNNs with fewer parameters may have longer inference times than Transformers with many more parameters. I agree with your earlier point that running-times are implementation dependent. However, a claim of faster-runtime/compute-efficiency needs to be empirically supported.*
> > > > >
> > > > > The point about the RNN architectures is correct, but two architectures based on the transformer encoder are being compared here, one of which has 6.903 million trainable parameters and the other 0.904 million.
> > > > >
> > > > > In order to carry out an empirical (but still contestable) comparison in terms of runtime, it would be necessary to run all comparative architectures individually on an isolated machine and GPU. Although this would be possible, it would require large resources and would call into question the cost-benefit factor.
> > > > >
> > > > > We thank the reviewer for the discussion, but regret that after this long debate there seems to be no convergence on any point.

---

> > > > > > ### Comment · Reviewer_hGyz · 2024-11-25
> > > > > >
> > > > > > My main concern with the evaluations is regarding fixed lookback-windows and performance on imputation task. I hope that this paper can be revised with extended evaluations --- e.g., longer lookback windows and more datasets for the imputation task. Hopefully, these evaluations can highlight the robustness of TSRM. Nevertheless, it would still be helpful to highlight the strengths and weaknesses of this method.
> > > > > >
> > > > > > Additionally, it would be helpful to optimize the hyperparameter search space. 150 configs for each forecast-horizon/dataset pair is not scalable in general.
> > > > > >
> > > > > > Thank you for engaging in a discussion with me. Good luck!

---

### Official Review · Reviewer_MWNg · 2024-11-04

**Soundness:** 3
**Presentation:** 2
**Contribution:** 2
**Rating:** 5
**Confidence:** 4

**Summary:**

The paper introduces a multilayered representation learning architecture called TSRM designed for multivariate time series forecasting and imputation. TSRM utilizes hierarchical encoding layers, each with a representation layer to capture diverse temporal patterns and an aggregation layer for combining learned representations. Based on a Transformer encoder-like configuration with self-attention mechanisms, TSRM outperforms SOTA approaches on seven benchmark datasets.

**Strengths:**

1. The article emphasizes model interpretability which is crucial for understanding and trusting its predictions.
2.  The article includes links to the source code, enabling researchers to reproduce the results.
3. The article demonstrates the model's effectiveness and generalization capabilities by testing it on datasets from various domains.

**Weaknesses:**

Generally, the technique of the manuscript is partially sound, and some major concerns are listed below:

1. The title of this paper could be clearer. It does not convey the specific techniques or strengths used, nor does it highlight the unique aspects of the work. The authors claim that this method can better handle different time scales, so perhaps emphasizing this could be beneficial. And this is just a suggestion.
2. The paper emphasizes that the targeted tasks are forecasting and imputation. However, the representation learning method in this paper does not specifically address forecasting and imputation, which may require additional clarification. Alternatively, future work could explore extending the method to other tasks.
3. There are too many subsections in the related work, leading to some redundant content. In particular, the "Few/zero-shot learning" subsection covers methods unrelated to this paper, so it could be appropriately condensed.

**Questions:**

Generally, the technique of the manuscript is partially sound, and some major concerns are listed below:
How is the kernel size chosen—is it based on specific criteria, such as the period length, or selected empirically? What about the sensitivity of this parameter? The results using $K$ independent 1D CNN layers with varying kernel sizes have not been compared to those with a uniform kernel size, even in the ablation study, which seems more important than discussing $N$.

---

> ### Author Response · Authors · 2024-11-18
>
> Thank you for your detailed review of our paper. We will go into the individual points in more detail below:
>
> > *The title of this paper could be clearer. It does not convey the specific techniques or strengths used, nor does it highlight the unique aspects of the work. The authors claim that this method can better handle different time scales, so perhaps emphasizing this could be beneficial. And this is just a suggestion.*
>
> Thank you for the suggestion. The intention was to give a short title, but you are right that the main message was lost in the process. We have changed the title to:
> TSRM: A lightweight architecture based on temporal feature encoding for time series forecasting and imputation.
>
> > *The paper emphasizes that the targeted tasks are forecasting and imputation. However, the representation learning method in this paper does not specifically address forecasting and imputation, which may require additional clarification. Alternatively, future work could explore extending the method to other tasks.*
>
> The term “representation learning” in this work refers to the learning and extraction of temporal features from the input time series. We are aware that there are other definitions of representation learning that go in the direction of pretraining and finetuning, such as PatchTST, which is then also used for a wider range of tasks. However, the initial focus of this work was to present and evaluate the architecture in the context of forecasting and imputation. Future work will address classification, anomaly detection, and clustering to the same extent, and preliminary experiments in this direction are already ongoing. We have included these points as an outlook in the conclusion of the manuscript.
>
> > *There are too many subsections in the related work, leading to some redundant content. In particular, the "Few/zero-shot learning" subsection covers methods unrelated to this paper, so it could be appropriately condensed.*
>
> It is true that parts of the related work section appear somewhat redundant in the current version. However, we consider it important to highlight the broad spectrum in the domain of time series analysis and to create a differentiation to our presented approach in order to clarify the context.
>
> Following the addition in the conclusion, the previously redundant areas in the related work are included in the future work part of the conclusion.

---

> ### Author Response · Authors · 2024-11-23
>
> We would like to thank you again for the review and would like to politely ask if there are any further questions or comments that we can help with.

---

> ### Comment · Reviewer_MWNg · 2024-11-26
>
> Thank you for the responses. However, it seems that the authors did not answer my question but only weaknesses. Nevertheless, this will not change my decision to maintain my current rating.

---

> > ### Author Response · Authors · 2024-11-26
> >
> > We apologize, the answer to your question got lost in the process.
> >
> > > *How is the kernel size chosen—is it based on specific criteria, such as the period length, or selected empirically?*
> >
> > The kernel sizes, just like the other CNN parameters in the representation layers, were part of the hyperparameter study (see Section A.4). In general, we found in the experiments that the configurations of the CNN layers are strongly dependent on the datasets, as shown in the configurations in Table 6.
> >
> > > *The results using independent 1D CNN layers with varying kernel sizes have not been compared to those with a uniform kernel size, even in the ablation study, which seems more important than discussing .*
> >
> > With the revision of November 18th, we added, among other things, experiments R1 (single CNN layer with kernel size 3) to the ablation study to show the effect of several independent ones compared to a single one. If we understand your question correctly, what you are missing in the ablation study is the comparison of several CNN layers of the same configuration compared to several CNN layers of different configuration. As described in Section 3.1, the different configurations serve the purpose of capturing different abstractions of the input time series, as is done, for example, with TimesNet [1]. The use of multiple CNN layers of the same configuration is not clear to us. Could you elaborate more on what benefit would be gained from supplementing our ablation study with the experiments you describe?
> >
> > [1] Wu, H., Hu, T., Liu, Y., Zhou, H., Wang, J., & Long, M. (2022). Timesnet: Temporal 2d-variation modeling for general time series analysis. arXiv preprint arXiv:2210.02186.

---

> > > ### Comment · Reviewer_MWNg · 2024-11-27
> > >
> > > Thank you for your response. You are correct in your understanding that I meant not using multiple CNN layers with the same configuration, but rather just one.

---

### Official Review · Reviewer_B9TQ · 2024-11-05

**Soundness:** 3
**Presentation:** 2
**Contribution:** 2
**Rating:** 5
**Confidence:** 4

**Summary:**

The paper introduces a novel architecture called Time Series Representation Model (TSRM) for multivariate time series forecasting and imputation. The TSRM architecture is based on a Transformer encoder-like configuration with hierarchically ordered encoding layers.  The TSRM architecture integrates representation learning within a multilayered model, where each layer features a distinct representation learning module. Each representation learning layer independently captures representations at a different hierarchical level, restoring the original input dimensions to enable hierarchical stacking of layers independent of the input dimension. The proposed approach achiever similar or better performance than state-of-the-art approaches on forecasting and imputation tasks while significantly reducing complexity in the form of learnable parameters.

**Strengths:**

- The paper focuses on an important problem of time series forecasting and imputation which can be useful in several practical use cases.
- Experiments show that the framework is able to outperform baselines on multiple datasets on the time series forecasting task.
- The paper is easy to follow.

**Weaknesses:**

- One major concern is the motivation behind certain architectural choices, such as the introduction of merge layers and the inclusion of representation layers in each hierarchical level. The ablation studies in Figure 2 show that removing the merge layers results in only a minor drop in performance  (0.379 -> 0.382 and 0.161 -> 0.165) compared to other ablation experiments, which raises questions about their necessity. Additionally, it is unclear whether having multiple representation layers is more effective than using a single set of CNN layers at the beginning of the Transformer architecture, as is common in computer vision. Without clear evidence of performance improvements, the novelty of the proposed approach appears marginal.
- Furthermore, the claim that each representation learning layer captures representations at a different hierarchical level requires qualitative analysis to support it.
- The paper's claim of achieving state-of-the-art performance on imputation tasks is not supported by the results in Table 2.
- How are masks handled in CNN layers (representation block)?
- Does the proposed approach need to be trained separately for different prediction horizons and different missing ratios or is it generalizable?

**Questions:**

Listed above in the weaknesses section.

---

> ### Author Response · Authors · 2024-11-18
>
> Thank you for your detailed review of our paper. We will go into the individual points in more detail below:
>
> > *One major concern is the motivation behind certain architectural choices [...]*
>
> We understand your point about the merge layer within our architecture and its apparently subordinate role in the performance improvement we show in our ablation study.
> However, in the experiments in the ablation study, the “no_merge” experiments did not remove the merge layer entirely, as the name might suggest, but merely disabled the gradients to prevent the merge layer from learning anything. The results show that learning merge layers plays a minor role in our architecture. However, the merge layers in our architecture fulfil the essential step of restoring the dimensional changes, which have been introduced by the representation layer before, to their original form to allow stacking of the encoding layer, independent of the input dimension. Accordingly, the merge layers are an essential part of our architecture, but you are right that their trainability does not lead to any major improvement in performance. We have changed the ablation study in relation to the merge layer to clarify this.
>
> > *Additionally, it is unclear whether having multiple representation layers is more effective than using a single set of CNN layers at the beginning of the Transformer architecture, as is common in computer vision. Without clear evidence of performance improvements, the novelty of the proposed approach appears marginal.*
>
> Unfortunately, we cannot follow your question as we do not use a transformer architecture, but only a modified version of the transformer encoder and build on a stacked architecture. Stacked CNN layers in front of a transformer is a different architecture and is therefore not part of our ablation study in which we investigate the effectiveness of individual elements of the architecture on performance.
>
> Or did you mean that the effectiveness of individual CNN layers in the representation layers should be studied separately from the other ablation experiments? We have added the experiments (R0 and R1) to the ablation study.
>
> > *Furthermore, the claim that each representation learning layer captures representations at a different hierarchical level requires qualitative analysis to support it.*
>
> You are right, there is an error in the manuscript. In our mansucript we wrote about hierarchically arranged representation layers, but we are actually talking about the encoding layers.
>
> The representation layers are not designed for hierarchical learning, but to encode the input at different levels of abstraction, as explained in Section 3.1. This process is similar to that of TimesNet, where the abstraction levels are determined by Fast Fourier Transformation [1], but differs in that we use static abstraction levels.
>
> The encoding layers, on the other hand, are stacked and independent of each other.
>
>
> >*The paper's claim of achieving state-of-the-art performance on imputation tasks is not supported by the results in Table 2.*
>
> You are right that this claim is not supported by the results we reported. For this reason we have formulated the second sentence of Section 4.3, paragraph "Results",  rather cautiously as follows:
>
> "*On the ECL and weather datasets, our architecture performs considerably well, while on the ETT datasets we were not able to match the current SOTA results."
>
> If you were referring to the abstract, we agree that one could indeed get the impression that we claim to achieve SOTA results for all tasks and datasets. We have changed the abstract in the manuscript to be more precise.
>
> > *How are masks handled in CNN layers (representation block)?*
>
> There are no masks in our architecture, such as those used in the vanilla Transformer. Masked values in the imputation task are replaced by the masking value -1 before they reach the architecture. This is explained in Section 4.3 within the second paragraph.
>
> > *Does the proposed approach need to be trained separately for different prediction horizons and different missing ratios or is it generalizable?*
>
> For the imputation task it is not necessary to retrain the model for different missing ratios as all layers are independent of the missing ratio.
>
> For the forecasting task, however, the model, or to be more precise, only the last layer (projection layer), would have to be retrained as this projects the input length $\times$ encoding size onto the prediction horizon.
>
> [1] Wu, H., Hu, T., Liu, Y., Zhou, H., Wang, J., & Long, M. (2022). Timesnet: Temporal 2d-variation modeling for general time series analysis. arXiv preprint arXiv:2210.02186.

---

> ### Author Response · Authors · 2024-11-23
>
> We would like to thank you again for the review and would like to politely ask if there are any further questions or comments that we can help with.

---

### Meta-Review · Area_Chair_XAXd · 2024-12-20

**Metareview:**

The manuscript introduces Time-series representational model that uses hierarchical encoding layer, each with a representational layer and aggregation layer, to capture diverse temporal patterns and combining the learned representations, respectively.

Weaknesses:
1. Although the above claim has been made in the manuscript, authors responded to reviewers that there is an error in the manuscript, where the hierarchically arranged representation layers, but we are actually talking about the encoding layers. The representation layers are not designed for hierarchical learning, but to encode the input at different levels of abstraction, as explained in Section 3.1. This process is similar to that of TimesNet, where the abstraction levels are determined by Fast Fourier Transformation [1], but differs in that we use static abstraction levels. [See response to Reviewer B9TQ].
2. In another response to the Reviewer B9TQ, authors concede that their results outsmarts SOTA on some data sets, while they do not in others. This is a point of contention, although they have revised the abstract to reflect this. Multiple reviewers have highlighted the lack of considerable performance gains by TRSM, in comparison to SOTA [Reviewer B9TQ and Reviewer hGyz].
3. Too many hyperparameters and tuning them to optimize results will be cumbersome.
3. As indicated by the reviewers, The manuscript will also need a lot of edits to avoid redundancies in the manuscript.

All in all, this is a good paper with novel method. However, the manuscript needs a thorough overhaul in terms of descriptions of the method and evaluation/ablations. We hope the authors find the review useful enough to revise and resubmit the manuscript to another venue.

**Additional Comments On Reviewer Discussion:**

Reviewers had pointed to flaws in the method description and evaluation results and authors conceded to the flaws during the rebuttal phase.

---

### Decision · Program_Chairs · 2025-01-22

Reject